# Canalized gene expression during development mediates caste differentiation in ants

Bitao Qiu ⬮ [1,2,11] ✉, Xueqin Dai [3,4,11], Panyi Li [5], Rasmus Stenbak Larsen [1], Ruyan Li [1], Alivia Lee Price [1], Guo Ding ⬮ [1,3], Michael James Texada ⬮ [6], Xiafang Zhang [3], Dashuang Zuo [3], Qionghua Gao [3,7], Wei Jiang [5], Tinggang Wen [5], Luigi Pontieri ⬮ [1], Chunxue Guo [5], Kim Rewitz ⬮ [6], Qiye Li ⬮ [5,8], Weiwei Liu ⬮ [3], Jacobus J. Boomsma ⬮ [2] ✉ and Guojie Zhang ⬮ [1,3,9,10] ✉

Ant colonies are higher-level organisms consisting of specialized reproductive and non-reproductive individuals that differentiate early in development, similar to germ–soma segregation in bilateral Metazoa. Analogous to diverging cell lines, developmental differentiation of individual ants has often been considered in epigenetic terms but the sets of genes that determine caste phenotypes throughout larval and pupal development remain unknown. Here, we reconstruct the individual developmental trajectories of two ant species, *Monomorium pharaonis* and *Acromyrmex echinatior*, after obtaining >1,400 whole-genome transcriptomes. Using a new backward prediction algorithm, we show that caste phenotypes can be accurately predicted by genome-wide transcriptome profiling. We find that caste differentiation is increasingly canalized from early development onwards, particularly in germline individuals (gynes/queens) and that the juvenile hormone signalling pathway plays a key role in this process by regulating body mass divergence between castes. We quantified gene-specific canalization levels and found that canalized genes with gyne/queen-biased expression were enriched for ovary and wing functions while canalized genes with worker-biased expression were enriched in brain and behavioural functions. Suppression in gyne larvae of *Freja*, a highly canalized gyne-biased ovary gene, disturbed pupal development by inducing non-adaptive intermediate phenotypes between gynes and workers. Our results are consistent with natural selection actively maintaining canalized caste phenotypes while securing robustness in the life cycle ontogeny of ant colonies.

In his *Ants* monograph, William Morton Wheeler concluded that there is a striking analogy "between the ant colony and the cell colony which constitutes the body of a Metazoan animal; and many of the laws that control the cellular origin, development, growth, reproduction and decay of the individual Metazoan, are seen to hold good also of the ant society regarded as an individual of a higher order"[1]. This century-old statement highlights putative parallels between irreversible major transitions to organismal multicellularity with a differentiated germline

and altruistic cellular soma on the one hand and to colonial superorganismality with physically differentiated queen and worker castes as higher-level germline and soma on the other hand[2]. It implies that, once cell fate or caste fate have been determined early in development, individual cells or ant larvae should follow analogous developmental trajectories that give rise to terminally specialized cell types or morphologically distinct adult caste phenotypes, respectively.

Some decades later, Conrad H. Waddington depicted metazoan embryogenesis as a pebble rolling downhill in a rugged epigenetic landscape of divergent valleys, with cells losing pluripotency as they, regardless of minor genetic or environment disturbances, commit to specific developmental trajectories, a process that he termed canalization[3,4]. In combination, these early insights suggest that there should be Waddington landscapes for ant colony development reflecting the analogous Wheelerian understanding of developmental processes at two hierarchical levels of organismality. More specifically, as (super) organismal differentiation proceeds, we expect individual cells or multicellular ants to increasingly commit to their target functional phenotype within the (super)organism by parallel transcriptomical and anatomical differentiation. Waddington also maintained that the degree of canalization is under natural selection and can be different across organs and tissues within metazoan organisms[3], so we expected to also find such differences among castes within superorganismal colonies. While molecular biology technology based on differential gene expression has now largely replaced Waddington's organismal perception of development, his diagrams remain instructive heuristic tools for analogous understanding of gene regulatory networks (GRNs) that affect cell differentiation in metazoan bodies and, as we conjecture here, caste differentiation in ant colonies.

Recent advances in single-cell transcriptomics have revealed many molecular details of Waddingtonian landscape differentiation[5–7], while reconstructing developmental trajectories in unprecedented detail and identifying key GRNs for cell fate determination[8]. However, no studies of comparable ambition have been pursued to track ontogenetic development of superorganismal colonies sensu Wheeler and quantify the canalization properties of caste differentiation. The phylogenetically diverse holometabolic ants, with their clear developmental stages, are particularly inviting to embark on such investigations but studies have so far used pooled samples or obtained rather few individual transcriptomes[9–11], which has precluded formal analyses of individual heterogeneity during the entire developmental process of caste differentiation. In particular, whether the sequence of larval and pupal caste differentiation is a canalized developmental process, in which specific genes initiate and regulate cascades of differential gene expression while shaping morphologically diverging phenotypes, is unknown.

We used low-input RNA sequencing to obtain >1,400 genome-wide individual transcriptomes covering the major developmental stages of the ants *Monomorium pharaonis* and *Acromyrmex echinatior*, while using *Drosophila melanogaster* for outgroup comparisons. These two ant species belong to the same subfamily, Myrmicinae, but differ in social and developmental characteristics. *M. pharaonis* is a highly polygynous (multiqueen) invasive ant with a monomorphic worker caste, where caste is known to be determined 'blastogenically' in early embryos before eggs hatch[12,13]. In contrast, the fungus-growing leaf-cutting ant *A. echinatior* has mostly single queen colonies but a polymorphic worker caste with small workers for brood nursing and gardening and large workers for foraging and colony defence, where caste determination occurs during early larval development[14,15]. All ants share the same common ancestor that evolved superorganismal colonies with specialized queens and lifetime unmated workers, with only a few cases of secondary loss of the queen caste being known[16]. The two ant species that we studied therefore probably represent typical models of colony and caste development found throughout most extant ants. We reconstructed developmental trajectories for gyne and worker caste differentiation via genome-wide gene expression profiling, using a novel algorithm for predicting caste phenotype before larvae express morphological differences. We then focused on the larval–pupal transition to quantify caste-specific canalization effects and their underlying pathways and we finally examined some of the key genes regulating caste phenotype canalization.

## Results

### A transcriptomic atlas for ant development

Developmental trajectory networks constructed from whole-genome transcriptomes (Fig. 1a and Extended Data Fig. 1; Methods) clustered individuals primarily by developmental stage and gradually also by caste phenotype. Adjacent developmental stages always grouped next to each other, as expected when development is a largely continuous process but we also observed distinct clusters in early embryonic stages (0–24 h) and for the late larvae to early pupae transition, indicating more discrete stage-specific transcriptomes. Principal component analysis (PCA) for the combined data from both species showed that the first axis separated the two ant species ($P < 0.0001$; two-sided $t$-test) while the second and third axes jointly separated individuals by developmental stage and caste identity ($P < 0.05$ for two-way analysis of variance (ANOVA) on the association between PC2 and developmental stage and caste; $P < 0.005$ for one-way ANOVA on the association between PC3 and developmental stage) (Fig. 1b).

Aligning developmental transcriptomes showed 67–81% similarity between the two ant species across the developmental stages (Fig. 1c), reflecting considerable conservation of developmental GRNs in the Myrmicinae subfamily to which both ant species belong. Developmental transcriptomes were more similar for gynes than for workers across the developmental stages, both when comparing the two ant species with each other and when contrasting the two *M. pharaonis* caste profiles with those of *D. melanogaster* females ($P < 0.01$ for all examined stages and for both comparisons; two-sided $t$-tests) (Fig. 1c). These patterns are consistent with gyne development being under stronger selection constraint than worker development across social insects with permanent caste differentiation[11]. However, it is important

**Fig. 1 | Homologous caste differentiation trajectories in two ant species.**
**a**, *M. pharaonis* developmental trajectories reconstructed from 568 individual transcriptomes covering all developmental stages and visualized with the Fruchterman–Reingold algorithm in an undirected network (Methods), so that individuals with similar transcriptomic profiles cluster together. Shading of connecting lines reflects the strength of mutual transcriptome correlations, ranging from 0.8 (light grey) to 1.0 (black). Symbol colours differentiate between embryos within eggs (white and blue), larvae (yellow), pupae (red) and adults (brown). Images courtesy of L.P.[18]. **b**, The first three PC axes for individual transcriptomes of *A. echinatior* and *M. pharaonis*, from first instar larvae to adults, constructed from 979 individual transcriptomes across the two ant species, using 7,838 one-to-one orthologous genes. Upper panel: probability density function (PDF) of PC1 values separating the two ant species. Lower panel: PC2 and PC3 jointly distinguish between developmental stages and caste phenotypes across individual transcriptomes, plotted separately for the two ant species. Lines connect the median PC values for each caste across development stages, showing that individuals follow very similar trajectories regardless of species identity. Symbol colours and shapes as in **a**. **c**, Between-species transcriptome similarities comparing *M. pharaonis* ($n_{gynes} = 171$; $n_{workers} = 154$) and *A. echinatior* ($n_{gynes} = 139$; $n_{workers} = 319$) (left) and *M. pharaonis* and *D. melanogaster* females ($n = 123$) (right), based on stage-specific Spearman correlation coefficients and plotted separately for gynes (red) and workers (blue). Between-species similarities peaked in third instar larvae where gynes and workers were similar but similarities were always lower in earlier and later stages, particularly for workers. The box plot shows the median (centre line), 25% and 75% quartiles (boxes), outermost values (whiskers) and data points (overlapping with boxes and whiskers).

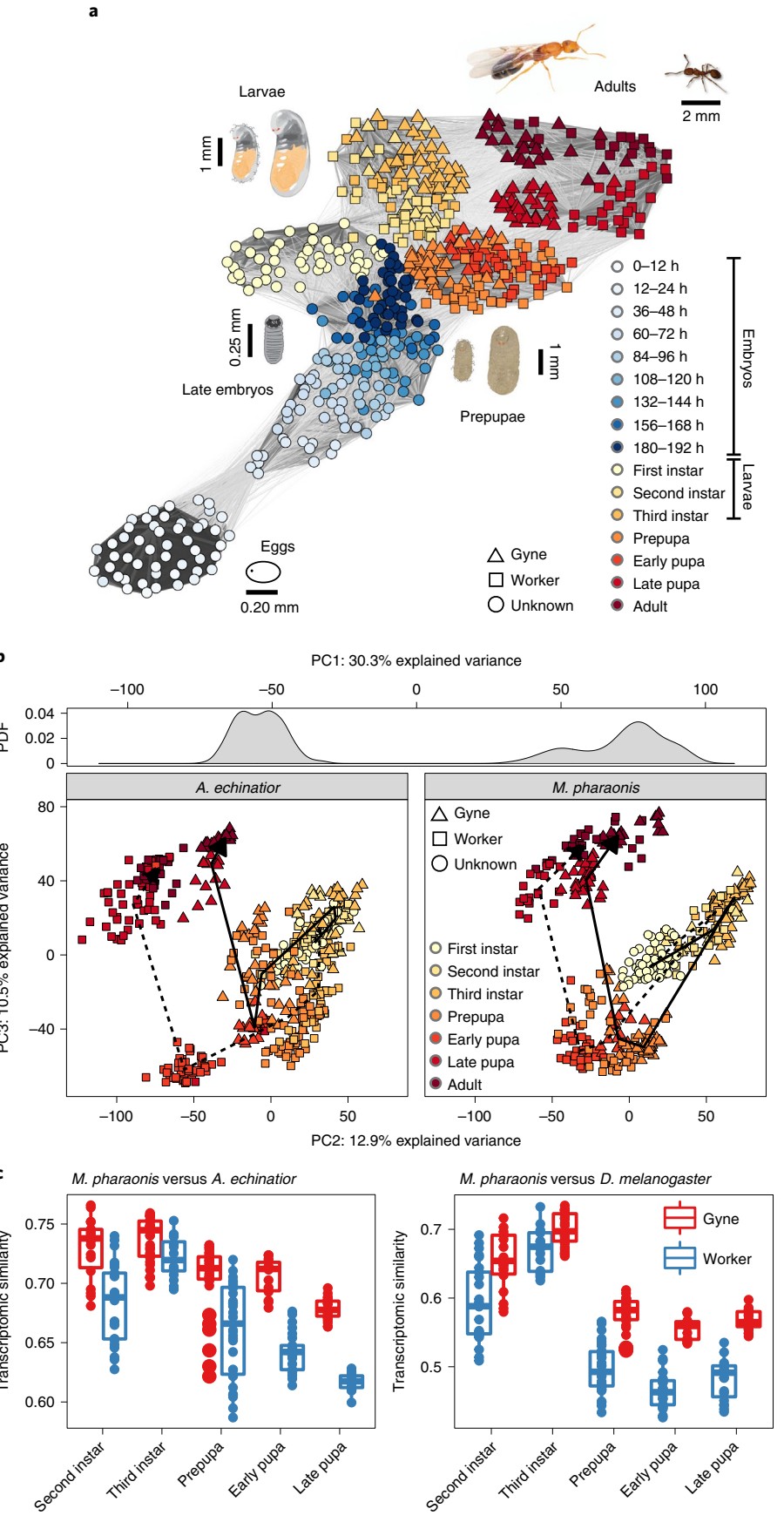

to acknowledge that gynes are elaborations of the ancestral reproductive phenotype of solitary female Hymenoptera, while differentiated worker castes are later innovations whose foundational GRNs evolved analogously, not homologously, in the superorganismal ants, bees and vespine wasps[17].

## Caste prediction in morphologically undifferentiated larvae

Morphological differences between gyne and worker individuals cannot be detected before the second and third larval instar in *M. pharaonis* and *A. echinatior*, respectively[14,18]. To identify individual caste phenotypes in earlier stages lacking morphological markers, we developed the backward progressives algorithm (BPA) that retrospectively infers the likelihood of individuals belonging to one caste or another (Methods; Extended Data Fig. 2a). BPA assumes that key genes active in the GRN at a specific stage should, albeit with modified expression, also participate in caste differentiation during the subsequent developmental stage, analogous to what is known for key transcription factors that specify cell types during metazoan development[5,8,19]. We validated BPA using embryonic sex differentiation data from *Drosophila* (Extended Data Fig. 2b) and confirmed the accuracy of BPA in samples of *M. pharaonis* larvae with known caste identity (Extended Data Fig. 2c).

We applied BPA to 54 transcriptomes of first instar *M. pharaonis* larvae (Fig. 2a) and predicted 12 of these to be reproductives (gynes and males) and 18 to be workers with >90% probability. We validated these predictions with RNA fluorescent in situ hybridization (HCR-FISH[20]) to assess the expression colocalization between *vasa*, a germline marker of first instar larvae and late embryos[21] (Extended Data Fig. 2d) and *LOC105839887* and *histone-lysine N-methyltransferase SMYD3* (*LOC105830671*). These two genes exhibit strong differential expression between predicted first instar caste phenotypes (Supplementary Table 1) and have binary gyne-worker expression in second instar larvae. First instar *LOC105839887* expression is visible in fat bodies while *SMYD3* colocalizes with *vasa* in the larval gonads (Fig. 2b, left panel). Both genes could be unambiguously detected in individuals with a *vasa*-specified germline and were always absent in individuals without a germline (Fig. 2b, right panel).

We also applied BPA in *A. echinatior* where we lacked morphological markers for second and first instar larvae. While third instar gyne larvae of this species can be unambiguously distinguished from worker larvae by their full-body curly hairs[14], this pilosity is not yet expressed in second instar larvae. BPA found that the first two PC axes constructed from the second instar transcriptomes separated second instar larvae by body size and third instar larvae by caste identities (Fig. 2c). Further inspection showed that the larger second instar larvae (suspected gynes) in fact have some gyne-like curly hairs in their ventral thorax region (Extended Data Fig. 2e), indicating that caste differentiation in *A. echinatior* begins before the second larval instar. To our knowledge, BPA is the first algorithm to achieve such accurate backwards predictions of developmental stages.

## Caste differentiation in ants is developmentally canalized

We next focused on the overall degree of canalization in genome-wide gene expression. For practical purposes, we defined transcriptome-level canalization as the statistical tendency for individual transcriptomes to start with a unimodal (pluripotent) distribution and gradually change to a bimodal (phenotypically committed) distribution with increasingly distinct peaks as development proceeds. For this purpose, we quantified the distributions of genome-wide developmental potential ($\Delta$) as a gyne or worker individual, using deviations in gene expression from average target profiles in subsequent developmental stages (Methods). The $\Delta$-values range between −1 and 1, with a positive value indicating that development in a target individual is gyne-biased and a negative value indicating worker-biased development. We found that the absolute $\Delta$-value between castes increased steadily in both ant species, while the variance of $\Delta$-values within castes, which we validated not

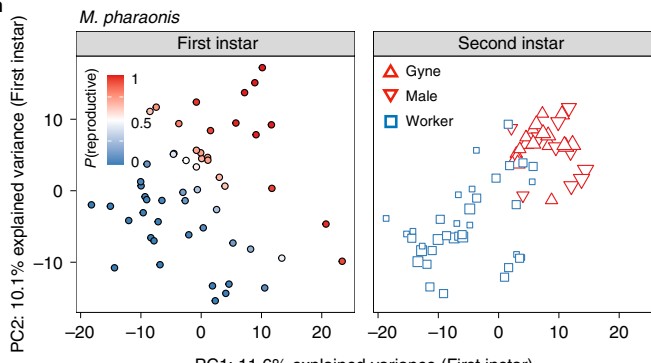

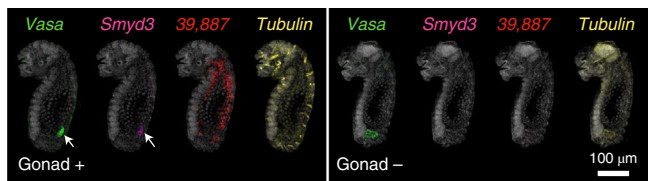

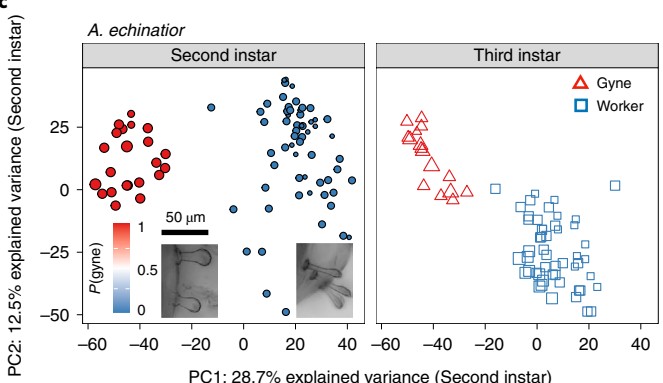

**Fig. 2 | BPA predicts individual caste phenotypes independent of external morphological traits. a**, BPA predictions of early caste identity in *M. pharaonis*, showing that the first two PC projections from first instar larvae (*n* = 54) match reproductives (an unknown mix of gynes and males) and workers among second instar larvae (*n* = 66). Colours of first instar sample symbols reflect the predicted probability to be a reproductive individual (left panel) and identify individuals with morphologically validated caste and sex in second instar larvae while also visualizing body length (right panel). **b**, HCR-FISH staining for two of the best predictor genes of caste in first instar *M. pharaonis* larvae based on presence (+) or absence (−) of gonad tissue, that is *LOC105839887* (red; expressed in fat bodies) and *Smyd3* (purple; expressed in gonads, white arrow), indicating that transcriptome-wide BPA assignments as gynes were correct because these two genes can only be detected in individuals with a germline (left Gonad+ panel). Germline presence was independently checked by *vasa* expression (green), showing that these transcripts were always undetectable in individuals without a germline (right Gonad− panel). The housekeeping gene *Tubulin* (yellow) was stained as a positive control (right images in both panels). **c**, BPA predictions of early caste identity in *A. echinatior* larvae, showing that the first two PC projections from second instar larvae (*n* = 84) matched the third instar individuals (*n* = 67) with known caste morphology. Individual samples are coloured according to their predicted probability of being gyne or worker in the second instar (left panel) while visualizing individual body lengths via the symbol diameters. Known gynes and workers in the third instar are represented by triangles and squares (right panel) confirming BPA segregation. The inserted epifluorescence microscope hair images refer to the ventral thorax region of a typical predicted second instar gyne (left) or worker (right).

to have been notably affected by technical artifacts (Supplementary information), became gradually reduced as development proceeds (both *P* < 0.0001; two-way ANOVA) (Fig. 3a and Extended Data Fig. 3).

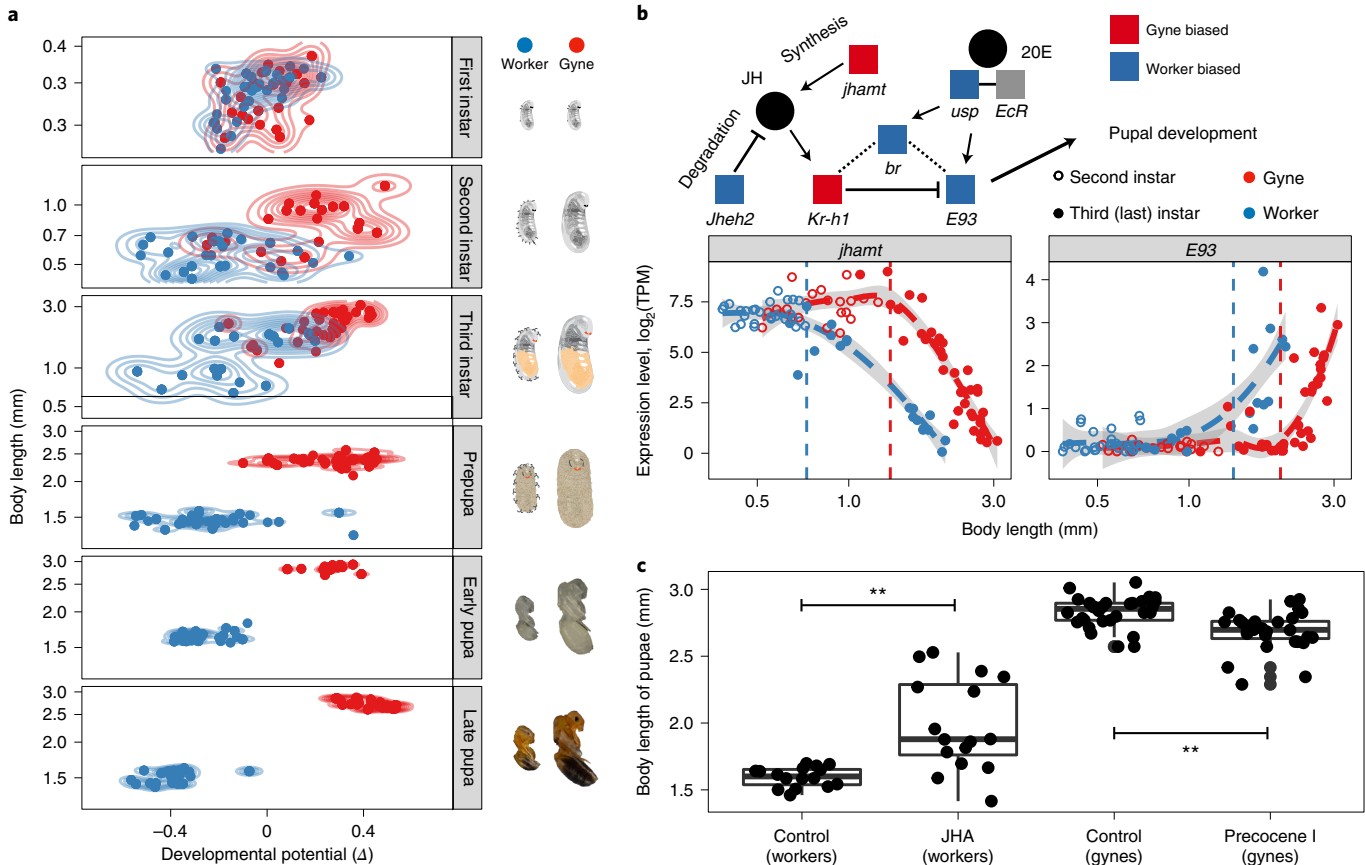

**Fig. 3 | Individual transcriptomes quantify caste canalization and its regulation by the JH signalling pathway. a**, Developmental potential scores ($-1 < \Delta < 1$) reflecting caste commitment for gyne (positive score) and worker (negative score) phenotypes across developmental stages of *M. pharaonis* as body length increases, corresponding to the representative images towards the right (courtesy L.P.). $\Delta$ values are based on normalized difference of transcriptomic distances between target individuals and an average gyne (calculated as the mean transcriptome across gynes within the same stage) and an average worker in the subsequent stage. A positive $\Delta$ value means that transcriptomic distance between a target individual and the average gyne in the next stage is less than between the target individual and the average worker in the next stage, so the target individual has a gyne-biased developmental potential. Caste identities were known from morphological characters, except for first instar larvae where they were inferred via BPA. **b**, Caste- and body length-specific expression of JH and ecdysone (20E) pathways, coloured according to

gyne-biased (red) or worker-biased (blue) expression in third instar larvae (upper panel). The expression of *jhamt* and *E93* showed both caste-specific (qualitative) and body size-specific (quantitative) thresholds (vertical dashed lines; estimated with a threshold regression model[57] (Methods)) (lower panel). Note that the downregulation of *jhamt* (terminating the JH biosynthesis) preceded the upregulation of *E93* to actively initiate metamorphosis (see Extended Data Fig. 5a for expression patterns of other genes in this pathway). **c**, In third instar larvae of intermediate body length, feeding with JHA increased worker pupal body length (two-sided *t*-test; $P = 0.0004$; d.f. = 16.38; $t = 4.44$; $n = 16$ for both control and treatment groups), while precocene I decreased gyne pupal body length (two-sided *t*-test; $P = 0.0001$; d.f. = 50.70; $t = -4.13$; $n = 30$ and 28 for control and treatment groups, respectively). The box plot shows the median (centre line), 25% and 75% quartiles (boxes), outermost values (whiskers) and data points (jitter plot overlapping with boxes and whiskers).

During this process, transcriptomic canalization in gynes was invariably stronger than in workers ($P < 0.05$ in all stages; two-sided *F*-tests), both in transcriptomic variation per se and in PCA patterns (Extended Data Fig. 3), indicating a higher degree of transcriptomic canalization in colony germline individuals.

## JH signalling regulates developmental caste canalization

Genome-wide transcriptomic canalization signatures amplified beyond the third instar when pupal metamorphosis starts (Fig. 3a), a critical stage in all holometabolous insects[22]. To understand the entirety of upstream regulation of caste differentiation, we used generalized linear models to account for the effect of larval body mass (Methods)[23] and identified 65 conserved genes with parallel gyne-worker bias that were associated with larval differentiation in both ant species (Supplementary Table 2). These early caste differentially expressed genes (DEGs) are significantly enriched for genes involved in fatty acid and hormone metabolism (Supplementary Table 3)—orthologues of these genes with gyne-biased expression are also highly expressed in the fat-body

tissues and tracheal system of *Drosophila* larvae (Extended Data Fig. 4a). In addition, multiple larval caste DEGs are associated with the juvenile hormone (JH) pathway, a key regulator of larval growth and molting in insects[22,24]. These included the genes *daywake*, encoding a haemolymph JH-binding protein, *LOC118646735*, a duplicate gene of *Drosophila Juvenile hormone acid O-methyltransferase* (*jhamt*) and *hexamerin*[25] (Extended Data Fig. 4b), confirming the important role of JH for caste differentiation[26,27].

We found that many genes involved in JH and ecdysone metabolism exhibited both body length-specific and caste-specific expression when larvae transition from the third instar to the prepupal stage (Fig. 3b and Extended Data Fig. 5a). In particular, the expression of *jhamt*, which delays the metamorphic molt[28], started decreasing in third instar worker larvae when body length exceeded 0.7 mm, while expression in gyne larvae did not decrease until larvae had reached twice that length of 1.4 mm (Methods) (Fig. 3b). A similar difference occurred in the expression of *Ecdysone-induced protein 93* (*E93*), a downstream transcription factor of the JH pathway for initiating metamorphosis[29]

(Fig. 3b), which started increasing when worker larval body length exceeded 1.4 mm but not in gyne larvae before they had reached 2.0 mm. These differences in body-length thresholds indicate heterochronic shifts between gyne and worker larvae in the JH signalling pathway, confirming the critical role of the JH pathway in regulating caste differentiation.

To experimentally prove the role of JH in caste canalization, we fed a JH analogue (JHA) to third instar worker larvae and precocene I, a JH inhibitor[30], to third instar gyne larvae of *M. pharaonis*. This showed that JHA significantly increased worker body length and that precocene I significantly reduced gyne body length (Fig. 3c). Furthermore, wing buds and simple eyes (ocelli), both typical gyne characters, were induced in the JHA-fed worker larvae (Extended Data Fig. 5c) while precocene I-treated gyne larvae had a significantly higher frequency of abnormal wings when they hatched as adults (P < 0.05 for Fisher's exact test; Extended Data Fig. 5d). Also these aberrant patterns of development confirmed that the JH signalling pathway is a key regulator for caste canalization in ants.

### Focal genes mediating canalization

To understand which genes are canalized during caste differentiation in the pupal stage, we developed a gene-specific canalization score ($C$) to track the developmental dynamics of between-caste gene expression divergence via the ratio of between-caste gene expression difference and stage-specific expression variance within castes (Methods). We identified 1,140 and 2,478 genes showing canalized expression in *M. pharaonis* (gynes versus workers) and *A. echinatior* (gynes versus small workers), respectively ($C > 3$ and $P < 0.05$ in one-sided Spearman's correlation tests) (Fig. 4a and Supplementary Table 4). Among these canalized genes, 457 showed the same caste-bias direction in both species, a significantly higher number than the background expectation of 88 ($P < 10^{-15}$; Fisher's exact test), indicating that gene-level canalization is evolutionarily conserved.

Comparison of the conserved canalized genes with tissue-level gene expression data in *Drosophila*[31] showed that gyne-biased canalized genes were highly expressed in ovaries, whereas worker-biased canalized genes were highly expressed in the brain and central nervous system (Extended Data Fig. 6a). Further analyses indicated that gyne-biased canalized genes were significantly enriched in flight muscle functions (for example, *tropomyosin-2* (*Tm2*), *troponin I* (*TnI*) and *troponin C* (*TnC*)) and female reproductive functions (for example, *T-complex protein 1* (*Tcp-1*), *krasavietz* (*kra*) and *merry-go-round* (*mgr*)) ($P < 0.05$ for both categories in hypergeometric tests). In contrast, worker-biased canalized genes were significantly enriched in neuronal and behavioural processes ($P < 0.05$ for both categories in hypergeometric tests), including *twin of eyeless* (*toy*), *hormone receptor 51* (*Hr51*) and octopamine receptors (*Octbeta1R* and *Octalpha2R*). These gene functions in a solitary insect are consistent with gyne caste specialization being targeted at dispersal and reproduction and worker caste specialization at more variable 'somatic' social tasks (Fig. 4a, Extended Data Fig. 6b and Supplementary Table 5).

### *Freja* as a crucial regulator of queen phenotype

The top gene with gyne-biased canalization in *M. pharaonis* is Hymenoptera-specific (Methods) and encodes a protein containing a predicted signal peptide and a leucine-rich repeat domain (Extended Data Fig. 7a). Caste-biased expression of this gene begins in second instar larvae and amplifies as development progresses (Extended Data Fig. 7b). In adult gynes, this gene is mainly expressed in the ovaries (Extended Data Fig. 7c) where its expression is restricted mostly to the ovarian follicle cells (Fig. 4b) which are essential for oogenesis[32]. We therefore named this gene *follicle related [gene-]expression in juvenile ants* (*Freja*, goddess of fertility in Old Norse).

We investigated *Freja*'s function through RNAi knock-down in late third instar gyne larvae. Relative to *GFP*-RNAi controls, adults of the *Freja*-RNAi treatment group had significantly reduced body and head size (two-sided *t*-tests; $P < 0.0001$ for both; Fig. 4c) and a higher frequency of abnormal wing morphology. As wings and large bodies relative to workers are unambiguous phenotypic markers of the adult reproductive (gyne/queen) caste in ants, we conclude that manipulation of *Freja* expression disturbed normal developmental canalization.

Because *Freja* remains highly expressed after pupal eclosion, we also manipulated *Freja* expression in adult *Monomorium* gynes with RNAi and examined effects on fertility. Both the size and the number of oocytes were significantly reduced in the *Freja*-RNAi group compared to the controls (two-sided *t*-tests; $P < 0.05$ for both; Fig. 4d and Extended Data Fig. 7e), indicating that *Freja*'s continued expression is crucial for gyne maturation and fertility after insemination. Thus, in addition to its necessary role in canalization of larval caste divergence, *Freja* maintains its differentiating functionality in adult gyne phenotypes. However, *Freja* was not a canalized gene in *A. echinatior*, a species where workers have retained ovaries to produce unfertilized (male) or inviable trophic eggs[33] (Extended Data Fig. 6).

## Discussion

We have shown that ant caste differentiation is canalized throughout development in a way that is remarkably analogous to the canalized development of metazoan cell lineages starting with the first cell divisions of a zygote, a potential functional similarity that was noted more than a century ago[34]. Also the deeper conservation of colony-level germline development is consistent with the secondary evolution of worker subcastes in ants being more common than the evolution of queen subcastes[35,36] and appears broadly convergent to what is known from development in animal bodies[5,6,37]. This suggests that ant colony-level ontogeny, unfolding as caste differentiation, has been maintained by consistent stabilizing selection, in spite of substantial modifications in the details of caste differentiation during the huge adaptive radiation of the ants. We also showed that the JH signalling pathway, known to be important for caste differentiation in social insects in general[26,38–41], mediates a heterochronic shift between gyne and worker larvae, suggesting that crucial metabolic changes control the regulation of both individual body size and the amplification of caste phenotype canalization.

Our study used an algorithmic approach (BPA) to predict caste phenotypes backwards in developmental time to identify gene markers before morphological caste differences emerge, a technique that should be broadly applicable in other kinds of developmental differentiation studies. Our analyses strongly suggest that caste in ants is determined by the interaction of body size and gene expression, rather than by one of these factors alone[42]. We covered the entire developmental process, starting with genome-wide transcriptomics and then zooming in on the JH and ecdysone pathways to finally focus on specific genes with key roles in canalization of caste development. Intriguingly, experimental inhibition of *Freja* finally showed that disruption of canalized genes results in non-adaptive intermediate phenotypes between gynes and workers, both early in development and in adults.

Our findings emphasize that caste-differentiated superorganismal colonies, as originally defined by Wheeler[34], are shaped by higher-level adaptations to predictably reproduce entire life-cycles. In this process, the complementary development of caste phenotypes requires coordination and buffering via expression changes of canalized genes, analogous to the dynamic regulation of cell differentiation during development of metazoan bodies[5,6,43,44]. We conjecture that obligate canalization of caste development will be a defining feature for all insect lineages that independently realized irreversible evolutionary transitions to superorganismality (ants, corbiculate bees, vespine wasps and higher termites)[2], setting the Wheeler-superorganisms—both annual and perennial—apart from their society-forming sister lineages[45]. However, our results indicate that even canalized developmental pathways can be changed when relevant genetic variation emerges and

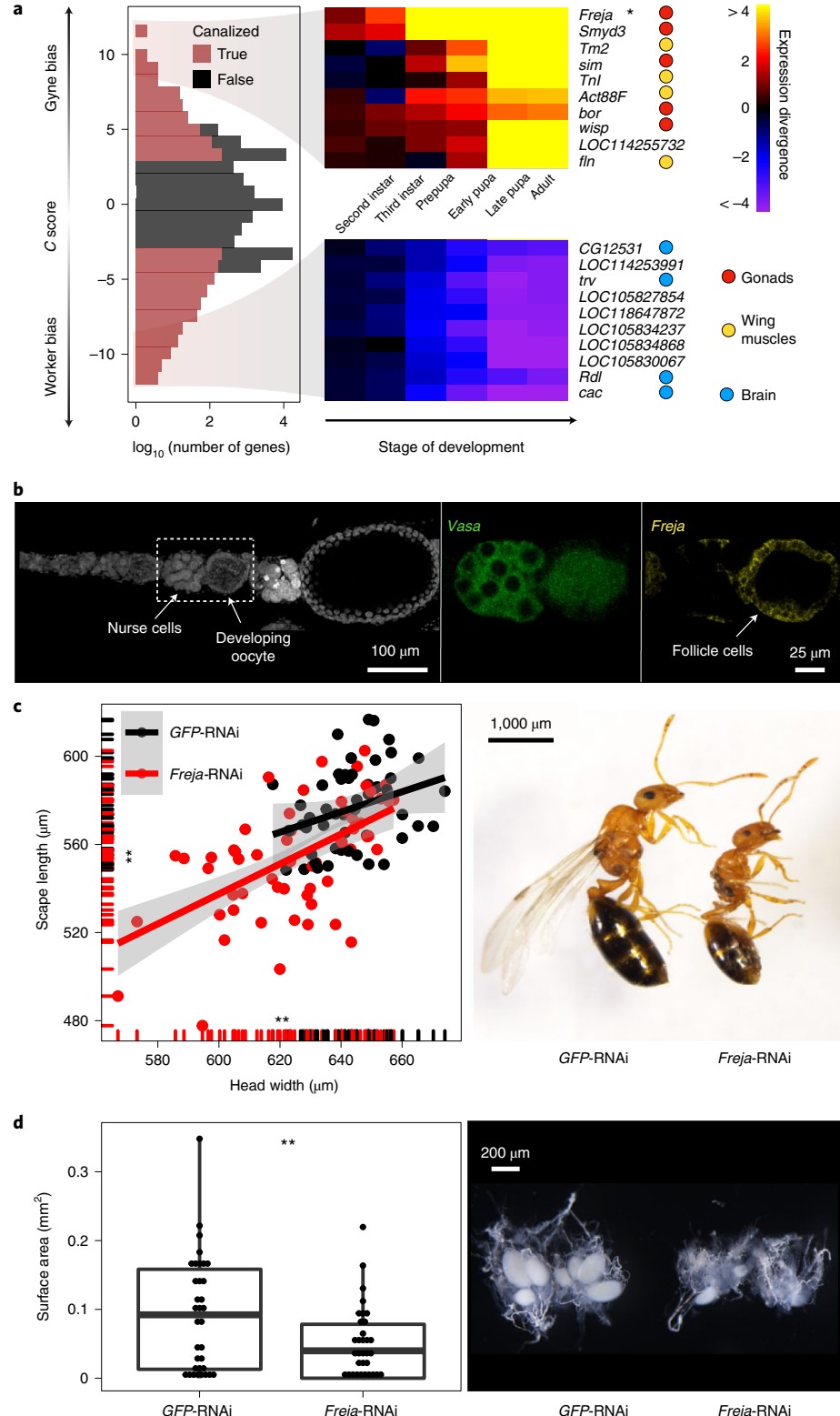

**Fig. 4 | *Freja* and other major caste canalizing genes in *M. pharaonis*. a**, Overall gene-specific canalization scores (left) with the ten most strongly gyne-biased (top centre) and worker-biased canalized genes (bottom centre) visualized in heatmaps and colour coded by their tissue expression in *D. melanogaster* (right). **b**, HCR-FISH staining of *vasa* (middle, green) and the top canalized gene, *Freja* (right, yellow) in an adult gyne ovariole (left), showing *Freja* expression in ovarian follicle cells. **c**, Compared to *GFP*-RNAi controls (black, *n* = 51), *Freja*-RNAi-treated third instar gyne larvae (red, *n* = 58) produced pupae with reduced antennal scape length and head width (two-sided *t*-tests; d.f. = 102.47 and 93.63

and *t* = −5.43 and −5.61, respectively; both *P* < 10⁻⁶; left) and more frequent abnormal wing development or complete lack of wings (16/66, 24.2%) compared to the control group (4/55, 7.3%) (*P* = 0.01; two-sided Fisher exact test; right). **d**, Compared to *GFP*-RNAi controls (*n* = 34), *Freja*-RNAi-treated adult gynes (*n* = 33) induced lower mass and number of oocytes (two-sided *t*-tests; d.f. = 55.31 and 57.51, *t* = −2.40 and −3.03 and *P* = 0.02 and 0.004, respectively; Extended Data Fig. 7e), impairing overall reproductive functionality. The box plot shows the median (centre line), 25% and 75% quartiles (boxes), outermost values (whiskers) and data points (overlapping with boxes and whiskers).

selection for directional change in caste phenotype is strong enough. This seems analogous once more with how cell lines diverge during adaptive radiation of multicellular organisms. Selection pressures thus appear broadly similar across domains of organizational complexity but the GRNs behind these convergent developmental processes will be uniquely different across lineages.

## Methods

### Experimental design

Ultralow-input transcriptome sequencing was performed for individual samples of two ant species, *M. pharaonis* and *A. echinatior* (both subfamily Myrmicinae), covering four larval stages, the prepupal stage, the early and late pupal stage and the adult stage (Extended Data Table 1). Individual transcriptomes were also obtained for nine embryonic stages (starting 12 h after egg laying and continuing sampling of older eggs with 12-h intervals) in *M. pharaonis*. In this ant species, caste is known to be determined 'blastogenically' in early embryos[12,13], unlike most other ants where caste phenotype is determined during larval development[46]. For later developmental stages where caste phenotypes were morphologically distinguishable (Extended Data Table 2), ~30 individuals per caste (gynes and workers in *M. pharaonis*; gynes, large workers and small workers in *A. echinatior*) were collected for each stage (Extended Data Table 1). For stages where caste phenotype could not be identified morphologically (before the second instar in *M. pharaonis* and before the third instar in *A. echinatior*), 30 eggs or 60 larvae per stage were collected to transcriptomically infer caste. Body length and head capsule width were also measured for all larval individuals to obtain another measure of developmental age that integrated with transcriptomic information during analyses. Reference individual transcriptomes for *D. melanogaster* were further obtained, covering development from newly laid eggs to newly eclosed adults, sampling ~20 eggs at 3-h intervals and ~20 larvae at 6-h intervals. See Supplementary information for details of sample collection, stage and caste identification, RNA and DNA extraction, transcriptome profiling and sample quality control.

### Testing RNA quality, constructing cDNA libraries and RNA sequencing

RNA quality testing, production of complementary DNA libraries and sequencing were performed at BGI, China. The quality of RNA samples was tested with Agilent 2100 Bioanalyzer and only samples with good RNA quality (RNA integrity number (RIN) > 4) were retained for RNA sequencing[47]. Ambion ERCC RNA Spike-In Mix (catalogue no. 4456740) was added to each sample according to the manufacturer's instructions before cDNA library construction, to be able to later verify the qualities of sequenced RNAseq data.

RNA sequences for each sample were first reverse transcribed into cDNA following the Smart-seq2 protocol[48], which was then randomly fragmented with Tn5 enzymes and linked with sequencing adaptors to obtain a complete cDNA library for each individual sample. Primers were then added to the cDNA libraries for PCR amplification and fragments ranging from 150 to 350 base pairs were selected for further cDNA circularization to construct sequencing libraries. The samples were then sequenced on a BGISEQ-500 platform using a 100 nucleotide paired-end ultralow-input RNA sequencing protocol.

### In situ hybridization chain reaction

**In situ probes.** Sequences for LOC105837931 (*Freja*) (XM_012683128.3), *vasa* (XM_012686851.3), LOC105839887 (XM_036293539.1), LOC105830671 (*Smyd3*) (XM_036287663.1), *actin5c* (XM_012666578.3), *tubulin* (XM_012685189.3) and *septin2* (XM_012667189.2) were downloaded from NCBI and provided to Molecular Instruments for probe set synthesis. Alexa Fluor 488 was used for the detection of *vasa*; Alexa Fluor 546 was used for the detection of *Freja*, *actin5c* and *tubulin*; Alexa Fluor 594 was used for the detection of LOC105839887; and Alexa Fluor 647 was used for the detection of *Smyd3* and *septin2*.

### RNA fluorescence in situ hybridization in *M. pharaonis* larvae and ovaries.

RNA fluorescence in situ hybridization (FISH) was performed following the whole-mount *Drosophila* HCR v.3.0 protocol[20] with some modifications in that *M. pharaonis* larvae and ovaries were fixed at room temperature in scintillation vials with 50% FPE (4% formaldehyde; 0.5× PBS; 25 mM EGTA) and 50% heptane. Fixation time was then adjusted so that ovaries were fixed for 30 min, first and second larval instars were fixed for 1–2 h and third larval instars were fixed for 12 h. Following fixation, the lower layer (FPE) was removed and replaced with methanol followed by vigorous shaking. The lower layer was replaced once more with methanol, at which point larvae and ovaries sink to the bottom of the vial. Larvae and ovaries were then dehydrated with several changes of methanol and stored at −20 °C. Proteinase K concentration and treatment time was adjusted to 30 µg ml$^{-1}$ for 7 min for first and second larval instars and 10 min for third larval instars and ovaries. Following amplification, one SSCT wash (5× SSC; 0.1% Tween-20; pH 7.0) was extended to 1 h with the addition of 4′,6-diamidino-2-phenylindole (DAPI; 1:1,000) for nuclear staining in first and second larval instars or overnight for third larval instars and ovaries.

FISH-stained larvae and ovaries were then transferred into increasing concentrations of glycerol in 5× SSC and mounted in Vectashield or 70% glycerol/ 30% 5× SSC for imaging. Images were captured on a Leica SP5-X inverted confocal laser scanning microscope. Image stacks were processed using Fiji/ImageJ[49].

### RNA interference experiments

Third instar reproductive (gyne or male) larvae were taken from nests and fixed with double-sided adhesive tapes. Interference double-strand RNA (dsRNA) was then injected into the upper ventral abdomen with a capillary needle (1B100F, World Precision Instrument), equipped on a micropipette puller (P-2000, Sutter) and Eppendorf FemtoJet injector (Femtojet 4i and Transferman 4r system, Eppendorf). For adult gynes, dsRNA was injected into the thorax, using the same equipment as in the larval injections, 5-, 7- and 10-d after they had eclosed from the pupal stage. For both larval and adult RNAi, 7,500 ng µl$^{-1}$ of dsRNA were used in both experimental (*Freja*) and control (*GFP*) groups. To improve the efficiency in larval injection, lipofectamine 2000 (11668019, Thermo Fisher) was added in *Freja* and *GFP* dsRNA liquid (dsRNA:lipofectamine 2000 was 3:1). The dsRNA was synthesized in vitro, following the instructions of the MEGAscript RNAi Kit (AM1626, Thermo Fisher).

The *Freja* T7 primer sequences for dsRNA synthesis were:
Forward: 5′-TAATACGACTCACTATAGGGCATCCATATCGTT GAAGGGC-3′
Reverse: 5′-TAATACGACTCACTATAGGGGTCCAGGTCGGT GAAGTTGT-3′
The *GFP* T7 primer sequences for dsRNA synthesis were:
Forward: 5′-TAATACGACTCACTATAGGGAGTGCTTCAGCCG CTACCC-3′
Reverse: 5′- TAATACGACTCACTATAGGGCATGCCGAGAGTG ATCCCG-3′

### Quantification of tissue-expression abundance with RT–qPCR

Heads, thoraxes and gasters (fourth and higher abdominal segments) were dissected from newly eclosed gynes and gasters were subdivided by tissue into digestive glands, cuticles, fat bodies and ovaries. Total messenger RNA of the dissected tissues was then isolated using TRIzol reagent (15596018, Thermo Fisher). Reverse transcription was performed using the PrimeScript RT reagent Kit with gDNA Eraser (RR047B, Takara and mRNA levels were quantified using TB Green Premix EX TaqTM II (Tli RNaseH Plus, RR820A, Takara) on a CFX96TM Real-Time system (BIO-RAD). Expression of *Freja* was normalized to the expression of *EF1α* in each sample.

The quantitative PCR with reverse transcription (RT–qPCR) primer sequences for *Freja* were:

Forward: 5′- AACAGGGCAAACTCAGATATTTAC-3′
Reverse: 5′- AGGCATCGATCGTTATCTCGG-3′
The RT−qPCR primer sequences for *EF1α* were:
Forward: 5′-TTCATTTATTGCTCTCACATCTACG-3′
Reverse: 5′- ACCGTTGCCCTTTCTACTCTAA-3′

## Quantification of ovary developmental status
Ovary developmental status was quantified by the number and the surface area of yolky oocytes. Ovaries were dissected and collected from 12-day-old gynes, where the number and the total surface area of yolky oocytes were counted and measured, respectively, from individual samples.

## JHA and precocene I feeding experiment
Third instar worker larvae of intermediate body length were treated with JHA (Methoprene, MCE HY-B1161; 5 mg ml$^{-1}$ in 10% EtOH PBS solution), while control worker larvae were fed with 10% EtOH PBS solution. Both 0.5 mg ml$^{-1}$ JHA and 10% EtOH were mixed with foods and offered on days 1, 3 and 8. To confirm the efficiency of JHA, the treated worker larvae were collected 24 h after day 1 feeding. After isolating total RNA, the expression of *Kr-h1*, a downstream gene of JH, was determined in both control and JHA groups. The expression of *Kr-h1* was normalized to *EF1a* in each sample. The RT−qPCR primer sequence for *Kr-h1* were:
Forward: 5′- AGGATATAACGCAGCTTCCTGT-3′
Reverse: 5′- GTGTGGCAGCGAACATTGTG-3′

Third instar gyne larvae with intermediate body length were treated with 1% precocene I (Sigma-195855; in 10% EtOH in PBS), while control gyne larvae were fed with 10% EtOH PBS solution. Both 1% precocene I and 10% EtOH were mixed with foods and offered on days 1, 3, 5 and 7.

To reveal the effects of JHA and precocene I on larval development, pupal stage samples were collected and body lengths were measured under a stereomicroscope (SMZ18, Nikon). For JHA and control groups, cohort percentages of pupation on days 10, 15 and 20 were also recorded to check whether development time was affected. When pupae eclosed into adults, their head width across the eyes and scape lengths (proxies of body size) were measured in the control and JHA groups using the same stereomicroscope.

## Constructing the developmental trajectory network
Within each species, the developmental trajectory network was constructed on the basis of the transcriptomic similarities among all samples. These similarities, as described in the Supplementary Information 'Sample quality control', formed a pair-wise similarity matrix among all samples. Values of this matrix were used to construct a weighted undirected network, where nodes represent samples and edges represent similar transcriptome profiles between connected samples. The weights of edges were measured by the Spearman's correlation coefficient among samples, indicating the level of transcriptome similarity. To increase the overall signal of the network, weak connecting edges (weight < 0.8) were removed and the threshold criterion is based on the empirical suggestion that correlation coefficients for anisogenic samples should be >0.8 (ref. [50]).

The weighted undirected network was then visualized with the force-directed layout algorithm using the igraph package (v.1.2.9) in R (ref. [51]). This algorithm takes the weights of edges into account, so that nodes with strong edges (samples with high transcriptome abundance similarities) are clustering together. For visualization purposes, edges' colour was set according to edges' weight, ranging from white (weight = 0.8) to black (weight = 1).

## Alignment of developmental transcriptomes and measurement of between-species transcriptomic similarity
Rate of development and number of instars differ between ant species and castes[1,52]. In our study species, there are three larval instars in

*M. pharaonis* and three to four in *A. echinatior*. Embryogenesis in *M. pharaonis* lasts for ~9 d, which is long compared to *D. melanogaster* where eggs hatch after ~24 h. Developmental stages thus need to be aligned between species before comparative analyses can be done.

To align developmental stages, either between castes or between species, developmental stage similarities were measured by their overall transcriptomic distance, calculated as 1 − Spearman's correlation coefficient of transcriptomes (or orthologous transcriptomes for cross-species comparison). Stages with the lowest overall transcriptomic distance (mean value for all same-stage samples) were assessed as the best-matched (aligned) stages and used for downstream between-caste and between-species comparisons.

Between-species transcriptomic similarity for each developmental stage was calculated as the mean value of 1 − Spearman's correlation coefficients among samples of the best-aligned stages, separately for each caste.

## The BPA
We developed a new algorithm that allows backwards prediction of caste identity on the basis of the sequential overall transcriptomic patterns. The algorithm compares (1) the transcriptomes of individuals at a target developmental stage with unknown caste identity and (2) the transcriptomes of individuals at the later stage where caste identity was known or had been assigned in a previous round. For each round of prediction, BPA performed four steps: normalization, feature selection, model training and prediction.

Because prediction data (of differentiation in unassigned transcriptomes of the target developmental stage) and training data (caste-assigned transcriptomes from the subsequent stage) represent two continuous developmental stages next to each other in time, developmental effects are expected to always contribute, but with quantitatively different effects, to both datasets. The first step of BPA (normalization) therefore removes such developmental effects by subtracting the mean expression levels and normalizing the expression variation across the two datasets, using the Combat package (from sva (v.3.40.0)) in R, which sets developmental stage as batch covariate[53]. The normalized transcriptomes thus represent expression levels that are independent of developmental stage and with a maximal likelihood to reflect segregation by caste in both datasets.

The second essential step of BPA is feature selection. This starts with a PCA of the prediction data, assuming that one or several of the PC axes from this dataset should be related to as-yet-unspecified caste identities in the target stage. We thus assumed that one or several of the top PC axes should include the best-possible set of caste PC axes driven by the expression difference between the caste phenotypes not yet identified. The second substep of feature selection is then to confront these PC axes with training data, by projecting them on the samples of the subsequent developmental stage. This comparison uses singular value decomposition to extract the coefficients of each PC axis and then multiplies them with the training data. This process produces new PC scores for each individual in the training data, which then allows the third substep of identifying the PC axes that best separate the known caste identities in the training data when performing ANOVA. The most significant PC axes are then assumed to be the shared caste PCs for both prediction and training data.

With the selected candidate feature (the best-fitting PC axes for caste) in place, the third and fourth steps of BPA are model training and prediction. We first applied linear discriminant analysis on the known caste PC values to train a predictive model for caste segregation. We then predicted individual caste identities of target stage individuals and assigned a probability of being gyne (reproductive of unspecified sex before the second larval instar) or worker to each individual. Once a complete round is completed, a prediction result for developmental stage *n* (S$_n$) can then be used to predict individual caste identities of transcriptomes at stage *n* − 1 (S$_{n-1}$), following the same four steps as in

the previous round of prediction, except that now the samples at $S_{n-1}$ were used as prediction data and the samples at $S_n$ as training data.

For BPA prediction of first instar larvae in *M. pharaonis*, body size independent caste DEGs of second instar larvae (number of genes = 173) were used for feature selection. Body size independent caste DEGs were identified from samples of these second instar larvae with approximately equal body lengths (ranging between 0.52 and 1.08 mm), using DESeq2 (ref. [54]) with the model: Exp ≈ caste + log(body length) (see section Detecting DEGs between castes). A gene was retained for feature selection if its adjusted *P* value was <0.05 and its $\log_2$ expression fold-change between castes was >0.5. Compared to using all genes, our use of caste DEGs for feature selection substantially increased the accuracy in our testing dataset (Extended Data Fig. 2c), probably because it removed the housekeeping genes whose expression is unrelated to caste differentiation. See Supplementary information for 'Validating the accuracy of BPA' and 'Testing the influence of sex on early developmental transcriptomes in *M. pharaonis*'.

## Quantification of transcriptome variation and difference

Within-stage transcriptome variation was measured by the mean value of 1 − Spearman's correlation coefficients between the transcriptomes of target samples and the transcriptomes of all other samples within the same stage, regardless caste identities for overall measurement or separately for each caste for within-caste measurement.

Between-caste transcriptomic differences were calculated as the average expression difference between gynes and workers (of the same stage) for all genes. The expression difference of a single gene was calculated as:

$$\mathrm{Dif}_{\mathrm{gene}} = \left| \overline{\mathrm{Exp}_{\mathrm{gyne}}} - \overline{\mathrm{Exp}_{\mathrm{worker}}} \right|$$

where the absolute value serves to remove the likelihood sign difference for worker or gyne bias.

As the gene expression levels were already normalized by $\log_2$, the expression difference between castes is equivalent to the absolute value of $\log_2$ expression fold-change between castes.

## Quantification of caste developmental potential

Developmental potential of target individuals was based on the transcriptomic distance between a target individual and its focal caste (measured as the average transcriptome of all same-caste individuals) at the subsequent developmental stage. If the transcriptomic distance between a target individual and a representative gyne was smaller than the distance between that target individual and a representative worker, the target individual was classified as being more likely to start developing into (or continue its development into) a gyne rather than a worker.

The mean transcriptomic differences between stages were first normalized by standardizing the expression level of each gene for all same-stage individuals. This step removed the quantitative differences between developmental stages (see first step (normalization) in The BPA section). The developmental potential ($\Delta$) of each individual ($i$) was then calculated with the following formula:

$$\Delta_{x,t} = \frac{\mathrm{dist}\,(i, \mathrm{worker}_{t+1}) - \mathrm{dist}\,(i, \mathrm{gyne}_{t+1})}{\mathrm{dist}\,(\mathrm{gyne}_{t+1}, \mathrm{worker}_{t+1})}$$

Here, dist ($i$, caste$_{t+1}$) is the transcriptomic distance between individual $i$ and the focal caste at the subsequent stage and is calculated as the Manhattan distance, a robust measure for transcriptomes that is commonly used for arithmetic calculations. The transcriptomic distance difference between castes was then normalized by dist (gyne$_{t+1}$, worker$_{t+1}$), which measures the transcriptomic distance between a focal gyne and a focal worker at the subsequence stage, so that $\Delta_{x,t}$ becomes a dimensionless measure. $\Delta_{x,t} = 1$ then indicates that individual

$i$ is equivalent to a representative gyne in the subsequent stage while $\Delta_{x,t} = -1$ indicates that the individual is equivalent to the representative worker.

## Quantification of gene-level canalization scores

Gene-level canalization scores were calculated based on the developmental dynamics of expression divergence between gyne and worker castes. For each developmental stage with known/predicted caste phenotypes, a modified *t* score for expression divergence of a target gene ($g$) was first calculated as:

$$t_g = \frac{\overline{\mathrm{Exp}_{\mathrm{gyne}}} - \overline{\mathrm{Exp}_{\mathrm{worker}}}}{s_{\mathrm{p}}}$$

Here, $s_{\mathrm{p}}$ is the pooled standard deviation for the expression levels of gynes and workers:

$$s_{\mathrm{p}} = \sqrt{S_{\mathrm{gyne}}^2 + S_{\mathrm{worker}}^2}$$

A high absolute value of a *t* score indicates a high between-caste expression difference or a low within-caste expression variance(s).

On the basis of the *t* scores across developmental stages, the canalization score ($C$) was then quantified to measure the developmental trend for the expression differences between castes. We defined the canalization score as:

$$C_g = -\log_{10}\left(P_g\right) \times t_{g,\times\mathrm{final}}$$

where $P_g$ is the *P* value of the correlation test for the absolute values of *t* scores across developmental stages and $t_{g,\mathrm{final}}$ is the *t* score of the late pupal stage when the morphological differentiation process between gynes and workers is largely completed. With the canalization score, we can thus capture the canalization level for each gene because a high value of $-\log_{10}(P_g)$ indicates an increasing between-caste difference or a decreasing within-caste variance across developmental stages and a high absolute value of $t_{g,\mathrm{final}}$ indicates a large between-caste expression divergence at the terminal stage.

On the basis of these canalization scores, we defined a gene as being canalized if $P_g$ was <0.05 and the absolute value of $C_g$ was >3.

## Identifying the phylogenetic origin of *Freja*

To investigate the phylogenetic origin of *Freja* (NCBI ID: LOC105837931), the orthologue group of this gene, that is, the set of genes (including both orthologues and paralogues) descended from a single gene in the last common ancestor, was identified (with Orthofinder) among 17 selected species, including 11 Hymenoptera and 6 species outside the Hymenoptera (Supplementary information on 'Detection of orthologues and homologues across species'). All gene members of the *Freja* orthologue group were found exclusively in the Hymenoptera, indicating that *Freja* is a hymenopteran order-specific gene.

Multiple sequence alignments were further performed on the protein sequences of the *Freja* orthologue group, using T-coffee (v.13.45.0) with default parameters [55]. With the aligned protein sequences, a gene tree was reconstructed, using IQ-TREE (v.2.1.4) with 1,000 replicates for bootstrapping and 1,000 replicates for Shimodaira–Hasegawa approximate likelihood ratio test [56].

## Body-length threshold regression model

To identify the threshold for a change in expression dynamics at a certain larval body size, gene expression levels of each caste were fitted with a continuous two-phase (segmented) model (M11) using the R package chngpt (v.2021.5-12) [57]. The threshold model can be expressed as:

$$\mathrm{Exp} \approx \alpha \times \mathrm{body\ size}_{\mathrm{before}} + \beta \times \mathrm{body\ size}_{\mathrm{after}}$$

Here, the threshold regression model detects a significant change of slopes ($\alpha$ and $\beta$) before and after a body size threshold and a threshold model is significant if the $P$ value for the log likelihood ratio test between the threshold model and the null model, Exp ≈ body size is < 0.05.

### Detecting DEGs between castes

Caste DEGs at each developmental stage were detected using the DESeq2 (v.1.32.0) package in R (ref. [54]). For all samples within each developmental stage, a read count matrix of transcriptomes (output from the Salmon mapping; see above) was loaded using tximport[58], which integrated expression profiles from the transcript level to the gene level. With these gene-level transcriptomes, DESeq2 was then used to model the expression level of each gene as: Exp ≈ caste and genes were defined as caste DEGs for a target developmental stage if their adjusted $P$ values were <0.05.

To reduce the confounding effect of body size on gene expression difference between castes (for example, second instar sexuals are always larger than second instar workers, so the expression difference might be the result of a larger body size)[23], body length measurements were further integrated to adjust slopes for the influence of body length and thus identify body size independent caste DEGs for second instar larvae. The expression level of each gene was modelled as:

$$\text{Exp} \approx \text{caste} + \log(\text{body length})$$

A gene was then assessed as a body size independent caste DEG if the adjusted $P$ value for caste was <0.05 and there was at least a 1.6-fold expression difference between castes. The expression difference between castes was estimated with a robust linear regression model[59].

### Determination of tissue origins of gene expression based on *Drosophila* database

Tissue origin of expression of target genes were based on the *D. melanogaster* gene expression atlas (FlyAtlas2)[31]. For larval stages, the expression data from larval main tissues, including brain, midgut, hindgut, Malpighian tubules, fat body, salivary gland and trachea, were used and their relative expression levels were calculated on the basis of their transcripts per million (TPM) values:

$$\text{RExp}_{\text{tissue}} = \frac{\log_2(\text{TPM}_{\text{tissue}} + 1)}{\sum(\log_2(\text{TPM}_{\text{tissue}} + 1))},$$

where one pseudo count was added to obtain a robust estimation in case of low TPM values.

For the pupal and adult stages, the expression data from adult female tissues, including brain, eye, thoracicoabdominal ganglion, midgut, hindgut, Malpighian tubules, fat body, salivary gland and ovary, were used. Relative expression levels were calculated as in the larval stages (see above).

### Reporting summary

Further information on research design is available in the Nature Research Reporting Summary linked to this article.

## Data availability

All transcriptomic data and the Whole Genome Shotgun project of *A. echinatior* are deposited at NCBI GEO and GenBank (BioProject Number: PRJNA767561).

## Code availability

Schematic diagrams and computational codes developed for this project and our in-house genome annotation for *A. echinatior* can be found at: https://github.com/BitaoQiu/devo-ants

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

## Acknowledgements

We thank the China National Genebank for providing transcriptomics sequencing service with MGI sequencers and Danish National Life Science Supercomputing Center, Computerome, for providing computational resources. We thank Z. Xiong for uploading the *A. echinatior* genomic data to NCBI. This work was supported by the Lundbeck Foundation (grant no. R190-2014-2827 to G.Z.), the Villum Foundation (Villum Investigator Grant, grant no. 25900 to G.Z.), the European Research Council (advanced grant no. 323085 to J.J.B.), the National Natural Science Foundation of China (grant no. 31820573 to G.Z.; no. 31501057 to Q.L. and no. 31900399 to W.L.).

## Author contributions

B.Q., J.J.B. and G.Z. conceived the project. B.Q., X.D., R.S.L., W.L. and G.Z. designed the experiments. B.Q. and G.Z. developed the bioinformatic methods. B.Q. analysed the data. X.D., P.L., R.S.L. and R.L. led experimental work, assisted by B.Q., A.L.P., G.D., M.J.T., X.Z.,

D.Z., Q.G., T.W., L.P. and L.W. B.Q. and P.L. managed the data, assisted by R.S.L., W.J. and C.G. K.R., Q.L., W.L. and G.Z. provided resources for the experiments. B.Q., J.J.B. and G.Z. prepared the manuscript, assisted by X.D., R.S.L., R.L., A.L.P. and M.J.T.

## Competing interests

The authors declare no competing interests.

## Additional information

**Extended data** is available for this paper at https://doi.org/10.1038/s41559-022-01884-y.

**Correspondence and requests for materials** should be addressed to Bitao Qiu, Jacobus J. Boomsma or Guojie Zhang.

[1]Villum Center for Biodiversity Genomics, Section for Ecology and Evolution, Department of Biology, University of Copenhagen, Copenhagen, Denmark. [2]Centre for Social Evolution, Section for Ecology and Evolution, Department of Biology, University of Copenhagen, Copenhagen, Denmark. [3]State Key Laboratory of Genetic Resources and Evolution, Kunming Institute of Zoology, Chinese Academy of Sciences, Kunming, China. [4]Kunming College of Life Science, University of Chinese Academy of Sciences, Kunming, China. [5]BGI–Shenzhen, Shenzhen, China. [6]Section for Cell and Neurobiology, Department of Biology, University of Copenhagen, Copenhagen, Denmark. [7]Guangxi Key Laboratory of Agric-Environment and Agric-Products Safety, College of Agriculture, Guangxi University, Nanning, China. [8]College of Life Sciences, University of Chinese Academy of Sciences, Beijing, China. [9]Evolutionary and Organismal Biology Research Center, School of Medicine, Zhejiang University, Hangzhou, China. [10]Women's Hospital, School of Medicine, Zhejiang University, Shangcheng District, Hangzhou, China. [11]These authors contributed equally: Bitao Qiu, Xueqin Dai. ✉e-mail: bitao.qiu.88@gmail.com; jjboomsma@bio.ku.dk; guojiezhang@zju.edu.cn

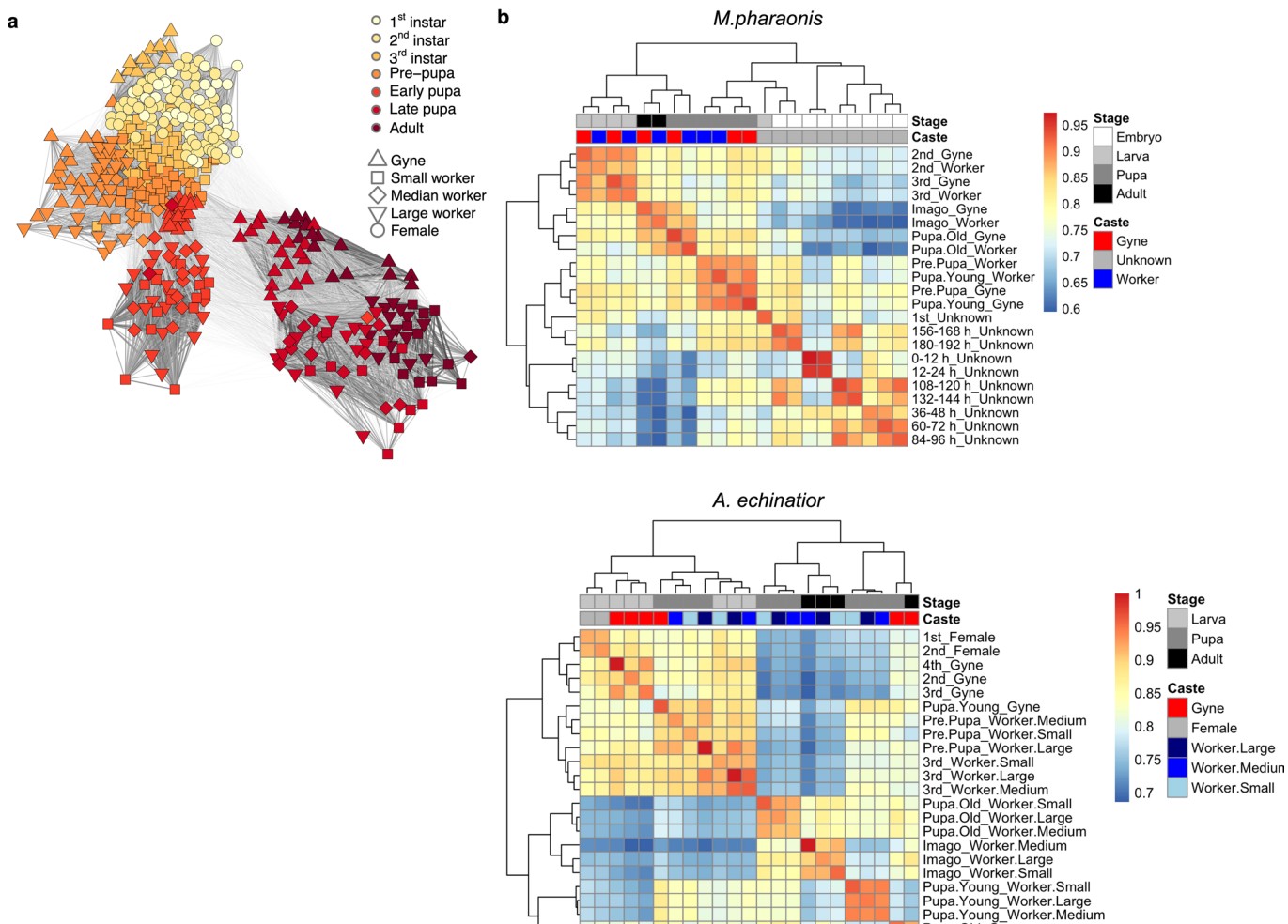

**Extended Data Fig. 1 | Developmental transcriptomes in ants.**
**a**, Transcriptomic developmental trajectories in *A. echinatior*, based on Spearman rank correlation similarity in gene expression across individual transcriptomes. Trajectories were constructed and visualized with a similar procedure as in Fig. 1a, except for *A. echinatior* having three worker subcastes, which exhibited increasing transcriptomic divergence across the pupal and adult stages. **b**, Between-stage transcriptomic similarity matrix in *M. pharaonis*

(upper panel) and *A. echinatior* (lower panel), based on the mean values of within-group and between-group correlation coefficients. In *M. pharaonis*, correlation coefficients within the same stage were always higher than between stages, and transcriptomic similarities consistently clustered by adjacent developmental stages. In *A. echinatior*, correlation coefficients within the same caste were higher than within the same stage, consistent with morphological caste differences being more substantial in *A. echinatior*.

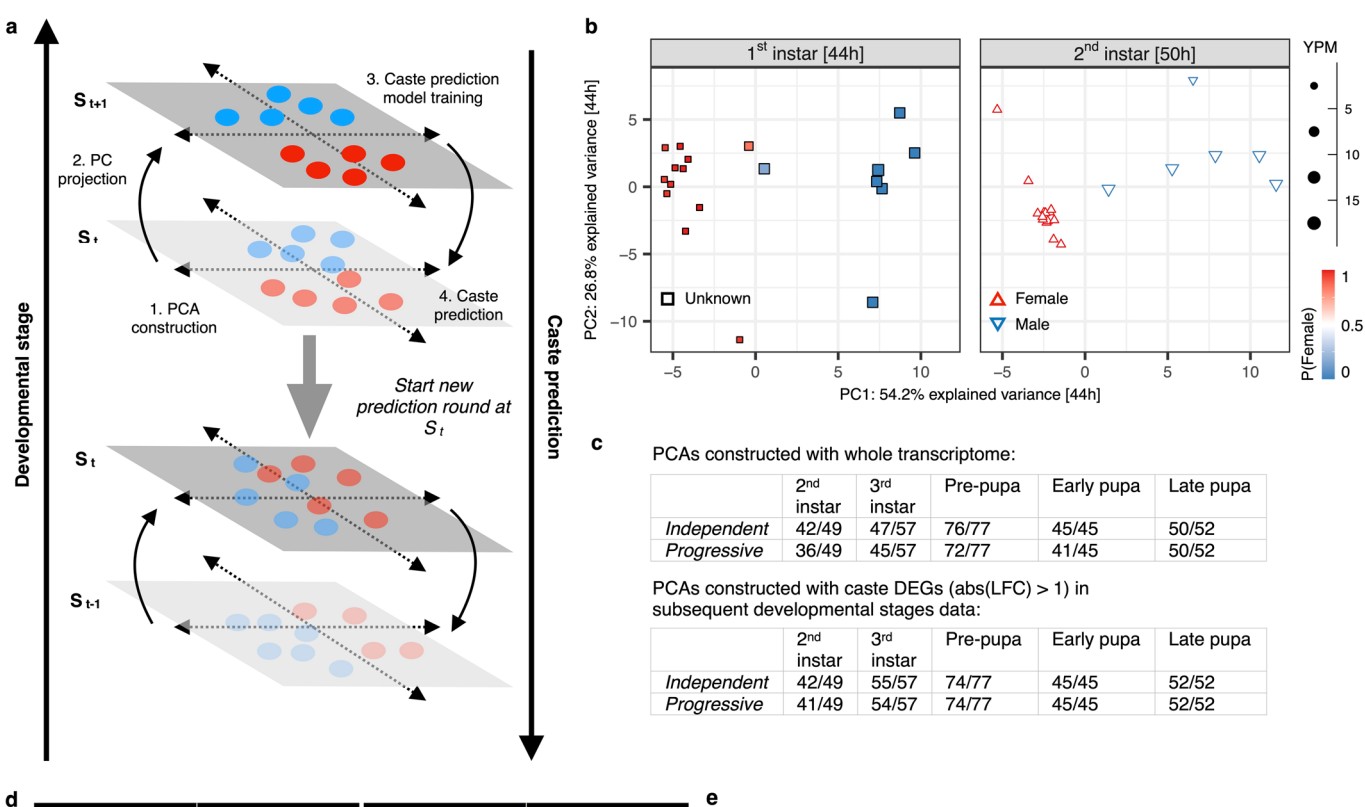

**c**

PCAs constructed with whole transcriptome:

| | 2nd instar | 3rd instar | Pre-pupa | Early pupa | Late pupa |
|---|---|---|---|---|---|
| *Independent* | 42/49 | 47/57 | 76/77 | 45/45 | 50/52 |
| *Progressive* | 36/49 | 45/57 | 72/77 | 41/45 | 50/52 |

PCAs constructed with caste DEGs (abs(LFC) > 1) in subsequent developmental stages data:

| | 2nd instar | 3rd instar | Pre-pupa | Early pupa | Late pupa |
|---|---|---|---|---|---|
| *Independent* | 42/49 | 55/57 | 74/77 | 45/45 | 52/52 |
| *Progressive* | 41/49 | 54/57 | 74/77 | 45/45 | 52/52 |

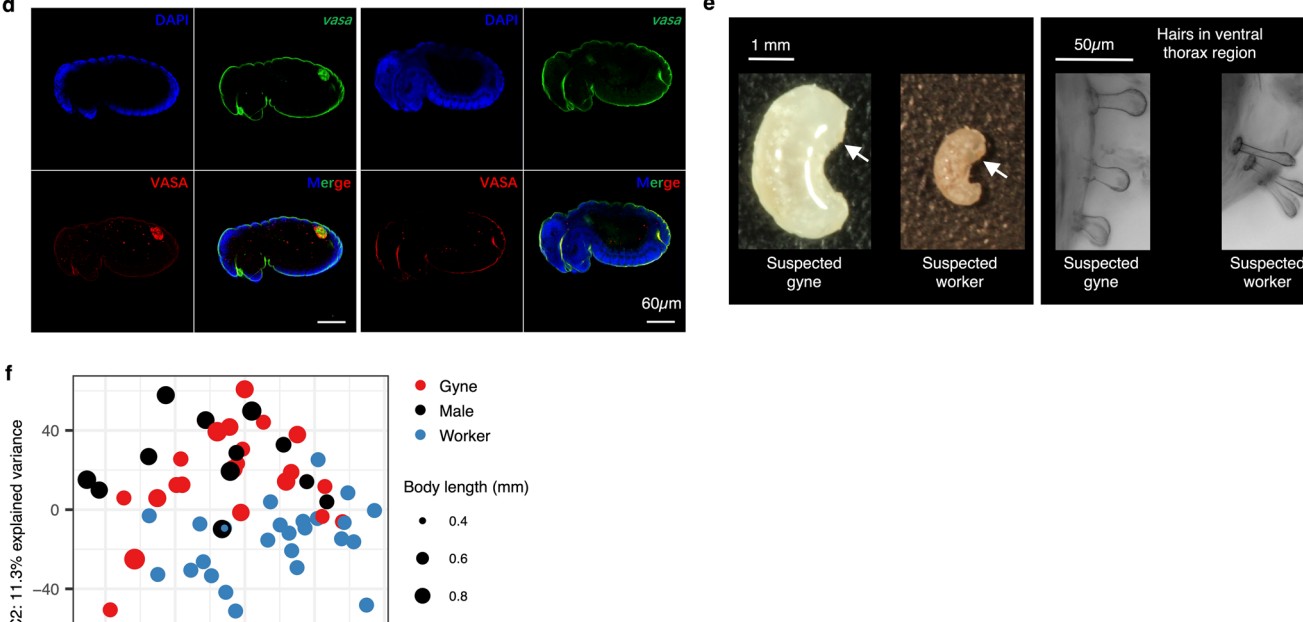

**Extended Data Fig. 2 | See next page for caption.**

**Extended Data Fig. 2 | Prediction of caste identities with BPA. a**, The *Backward Progressives Algorithm* (BPA) predicts caste identities in previous developmental stages, using non-validated transcriptomes at the target stage ($S_t$) to construct PCAs and then projects these onto the subsequent stage ($S_{t+1}$) where caste identities are known. The BPA then uses the known caste labels at $S_{t+1}$ to identify PC axes that are associated with confirmed caste identity and uses linear discriminant analysis to train a predictive model, assuming that the PC axes at $S_{t+1}$ are also associated with caste identities at $S_t$ as expected under developmental continuity. The trained model then predicts caste identities at $S_t$, after which it assumes these predicted caste identities to be real and initiates a next round to predict caste identities at stage $S_{t-1}$. This process continues until the prediction likelihoods at $S_{t-n}$ become too low to be informative. **b**, BPA predicted the sex of sampled individuals among 1st instar (44 hour) larvae in *D. melanogaster* (left panel). While sex can be distinguished in 2nd instar (50 hour) larvae (right panel) via genotyping after simultaneous DNA and RNA extraction, the biomass of 1st instar larvae was too small to perform such simultaneous extractions. By examining the proportion of reads that mapped to the Y chromosome of *Drosophila*, we found that predicted males in 1st instar larvae had a significantly higher proportion of reads mapped to the Y chromosome, confirming our prediction. Individual samples are coloured according to their predicted probability to be female or male in the 1st-instar and symbols were sized according to the number of reads mapped to Y chromosome per million reads (YPM). **c**, Validating BPA on samples of developmental stages with known caste identities used individual *M. pharaonis* transcriptomes of individuals with distinct morphology. The table presents prediction accuracies for each target stage, calculated as the ratio of the number of correctly assigned individuals and the total number of individuals sampled at each stage, using two alternative approaches: 1. *Independent*: Predicting caste at each targeted stage ($S_n$) using the observed (true) caste labels as training stage ($S_{n+1}$) to examine the prediction accuracy when the training caste identities are in fact known from morphological information. Here, the ratio in each stage reflects the accuracy of BPA in each stage. *Progressive*: Predicting caste starts from the late pupal stage using adults of known caste identity as training data. Here, BPA is then performed progressively using the predicted caste labels in late pupae to predict the caste identity in early pupae. This process was repeated recurrently until the 2nd larval instar. As the first step of BPA constructs a PCA from target stage data, we also compared the accuracies between PCAs obtained from whole transcriptomes and PCAs obtained from caste DEGs at the subsequent stage (training data). We achieved a higher prediction accuracy when PCAs were constructed with caste DEGs at the subsequent stage compared to using whole transcriptome PCAs, probably because the DEG method excluded uninformative housekeeping genes. **d**, Anti-body staining (VASA protein, red. RRID: AB_2893405) and in situ hybridization (*vasa* RNA, green) in 192-hour old embryos of *M. pharaonis*, showing that germline differentiation has already occurred at this stage. Among the 67 examined embryos, 18 (27%) could be documented to have no germline, indicating that it should be possible to match these presence/absence results among 192-hour embryos with BPA predictions based on 1st instar larval transcriptomes. **e**, Second instar larvae of *A. echinatior* lack the full-body curly hairs that distinguish gynes from workers in the 3rd larval instar, which means caste cannot be identified morphologically. We applied BPA to predict caste identities among 2nd-instar larvae (Fig. 2a). A closer inspection showed that suspected gynes have in fact some gyne-like curly hairs, which are thicker than those in suspected workers, on their ventral thorax (arrows). These observations indicated these individuals are future gynes and were consistent with our BPA predictions. **f**, Among 2nd instar larvae of *M. pharaonis*, PCA with whole transcriptomes showed that the overall transcriptomic difference between gynes and males was not significant ($P = 0.28$) while reproductive larvae of both sexes were always separated from worker larvae ($P < 5e-5$ and $P < 5e-6$ for gynes and males, respectively). This is consistent with images of 2nd instar gyne and male larvae being indistinguishable after we used microsatellite genotyping to determine whether individuals were haploid (male) or diploid (female). Numbers of differentially expressed genes (adjusted $P$ value < 0.05, detected with a generalized linear model that accounted for body size differences) also support this conclusion: 152 genes were differentially expressed between gyne and worker larvae, while 50 genes were differentially expressed between gyne and male larvae.

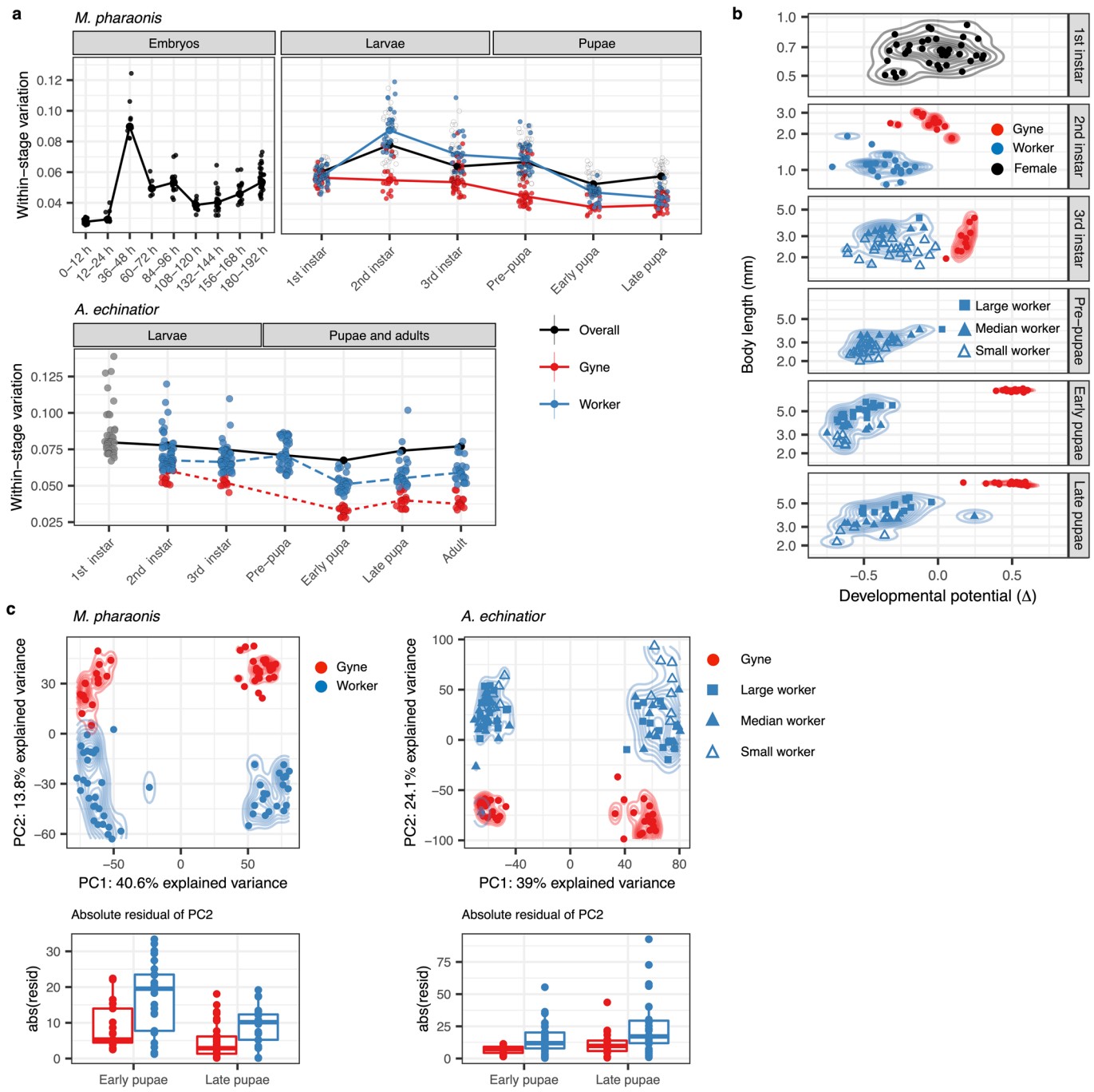

**Extended Data Fig. 3 | See next page for caption.**

**Extended Data Fig. 3 | Transcriptomic canalization during caste differentiation in ants. a**, Within-stage transcriptome variation in *M. pharaonis* (upper panel) from 0–12 h old embryos to late pupae, plotted separately for gynes (red), workers (blue) and all individuals within each stage (black) depending on available information. The lower panel gives the same information for *A. echinatior*, where embryonic data were not available and 1st instar caste phenotypes (grey) were inseparable with BPA. Transcriptome variation was quantified as $1 - r$, the extent of imperfection of transcriptome-level Spearman's correlations between a target individual and all other same-stage and same-caste individuals. Caste identities of 1st instar individuals of *M. pharaonis* and 2nd instar individuals of *A. echinatior* were predicted by BPA. In *M. pharaonis*, transcriptome variation for all individuals peaked in 36–48 h old embryos (equivalent to the gastrulation stage, 6–7, in *Drosophila* larvae). For both species, transcriptome variation among gynes was consistently lower than among workers. In pupal stages of *M. pharaonis*, transcriptome variation across all individuals exceeded transcriptome variation for the gyne and worker subsets, indicating that transcriptome differences primarily reflected realized caste differentiation, in contrast to the pattern observed across the larval stages, where the black curve was intermediate between the red and blue curves. Fourth larval instar and prepupal gyne samples of *A. echinatior* were excluded from this analysis, because these samples were sequenced in a different technical batch, making their transcriptome variation incomparable with the other samples. **b**, Developmental potential ($\Delta$) for individual gynes and workers in *A. echinatior*, measured as the transcriptomic distance between a focal individual

and an average gyne or worker (pooling all three worker subcastes) phenotype in the next developmental stage. Developmental potential was quantified and presented as in *M. pharaonis* (Fig. 3a), except that all three worker subcastes were included. Caste identities for gynes and (pooled) workers in 2nd instar larvae were predicted by BPA. As in the **panel a**, fourth larval instar and prepupal gyne samples were excluded to avoid a batch effect. **c**, PCAs of early and late pupal stage transcriptomes in *M. pharaonis* (early pupa gynes, $n = 17$; early pupa workers, $n = 28$; late pupa gynes, $n = 30$; late pupa workers, $n = 22$) (left) and *A. echinatior* (early pupa gynes, $n = 18$; early pupa workers, $n = 47$; late pupa gynes, $n = 18$; late pupa workers, $n = 42$) (right). For both species, the first PC axis (PC1) separates individual transcriptomes by developmental stage (early pupae to the left and late pupae to the right) while PC2 captures the caste-related transcriptomic variation. The overall transcriptomic difference between gynes (red) and workers (blue) increases from the early to the late pupal stage (upper panels), and the absolute values of the PC2 residuals (lower panels), representing the variation within each caste, were always lower among gynes than among workers ($P < 1 \times 10^{-3}$ for both species, two-sided *t*-tests). This is consistent with the mean extent of canalization being stronger in gynes than in workers. In *A. echinatior*, the absolute residual differences increase for the workers, consistent with *A. echinatior* having worker subcastes that differentiate rather late in development. Box plots show the median (centre line), 25% and 75% quartiles (boxes), outermost values (whiskers) and data points (overlapping with box and whiskers).

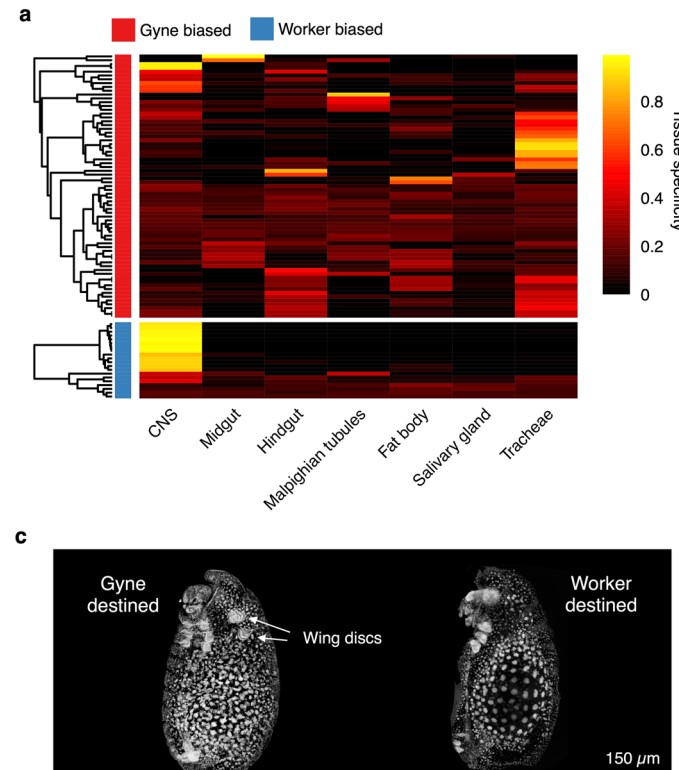

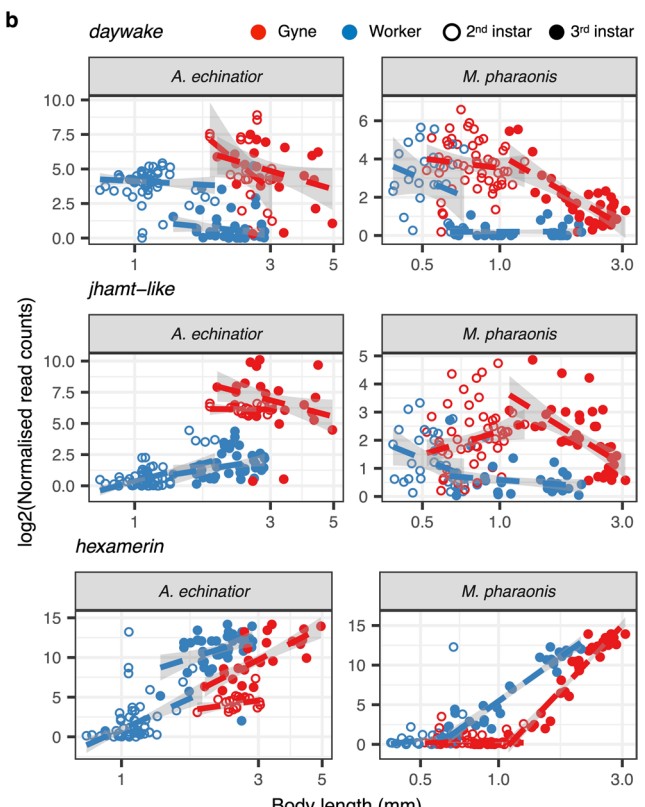

**Extended Data Fig. 4 | Early larval caste differentiation in ants. a**, Tissue-specific relative expression levels for the conserved caste-biased DEGs in early larvae, shown separately for gyne-biased (rows marked in red) and worker-biased (blue) genes. Heatmap brightness of cells reflects tissue specificity, the percentage of transcripts from targeted tissues (columns), ranging from 0% (black) to 100% (yellow). These relative abundances, based on the larval gene expression atlas of *Drosophila*, show that the gyne-biased DEGs in the early larval stages were mainly expressed in the midgut, fat body, and tracheal tissues, while the worker-biased DEGs were mainly expressed in the brain and central nervous system. **b**, Expression profiles of *circadian clock-controlled protein* (*daywake*),

*juvenile hormone acid O-methyltransferase-like* (*jhamt-like*) and *hexamerin* among gynes and workers of the two ant species as larvae grow. All three genes are associated with the juvenile hormone signalling pathway and are significantly differentially expressed between castes in 2nd and 3rd instar larvae. Expression profiles are plotted against body length (log scale) to show expression dynamics as larvae grow in body length. **c**, DAPI staining of a representative early 3rd instar worker larva and a representative 2nd instar gyne larva of *M. pharaonis*. These animals display similar body size but wing discs (arrows) were only visible in the gyne larvae, indicating that caste determination and differentiation has already been initiated well before this early larval stage.

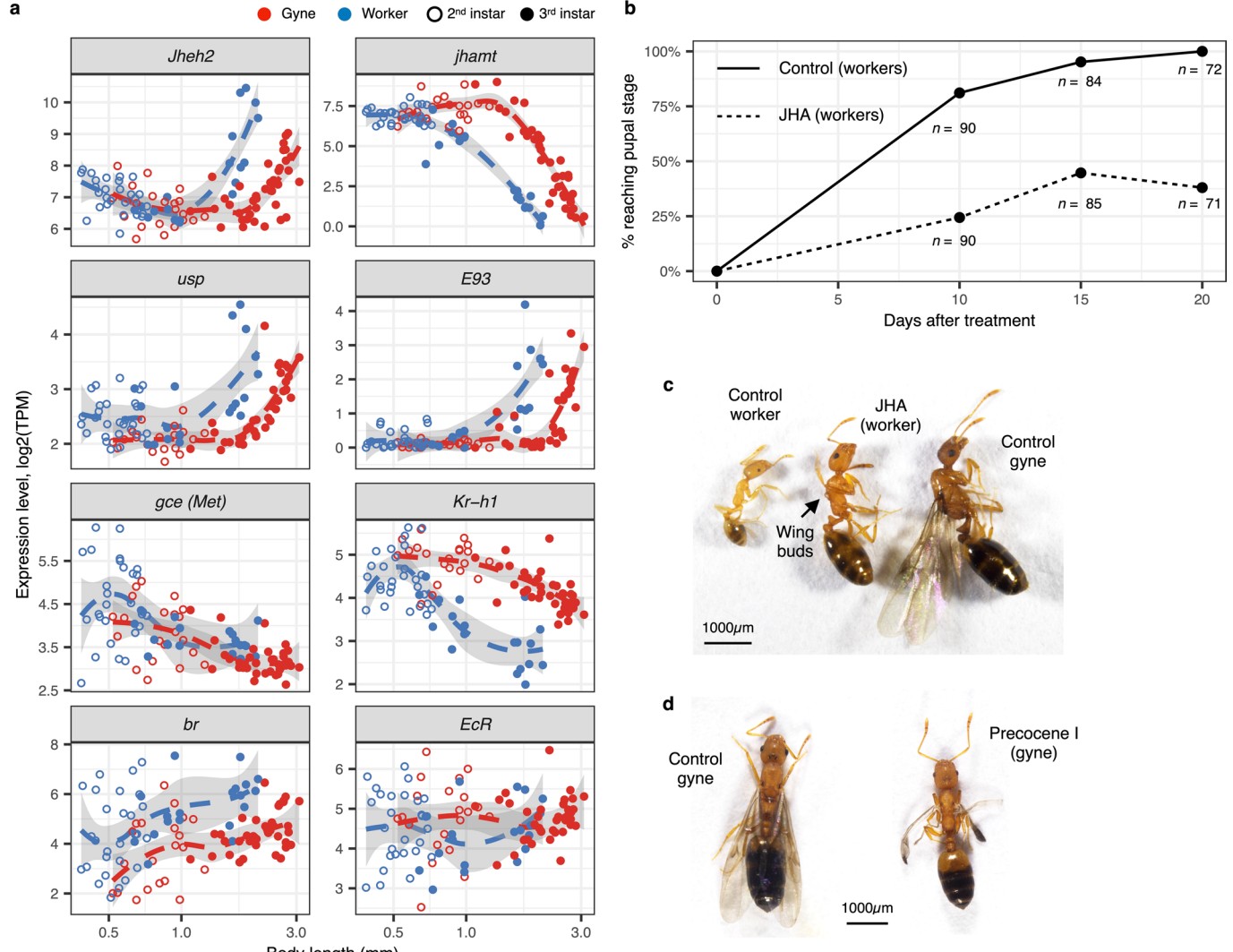

**Extended Data Fig. 5 | JH and E20 signalling pathways play key role in the regulation of canalized caste phenotypes. a**, Expression profiles of eight key regulators for insect metamorphosis that are part of the juvenile hormone and ecdysone signalling pathways (Fig. 3b), plotted against body length (log scale) of 2nd and 3rd instar *M. pharaonis* larvae. The expression levels of half of these genes (*jheh2, jhamt, usp* and *E93*) showed caste-specific body length thresholds in the 3rd larval instar. This pattern indicates gyne and worker individuals are gated by different critical masses for entering the metamorphic molt. **b**, Compared to the control group (3rd instar worker larvae fed with 10% EtOH

PBS), feeding JH analogue (JHA) to 3rd instar worker larvae delayed achieving pupation. **c**, JHA-fed 3rd instar worker larvae induced inter-caste with phenotype intermediate between gyne and worker. JHA-fed workers have larger body size and developed wing buds (arrowed), however, they never developed ovaries (not show), indicating early bifurcation between colony germ–soma phenotypes. **d**, Compared to the control group (3rd instar gyne larvae fed with 10% EtOH PBS), precocene I fed 3rd instar gyne larvae were smaller and developed abnormal wings.

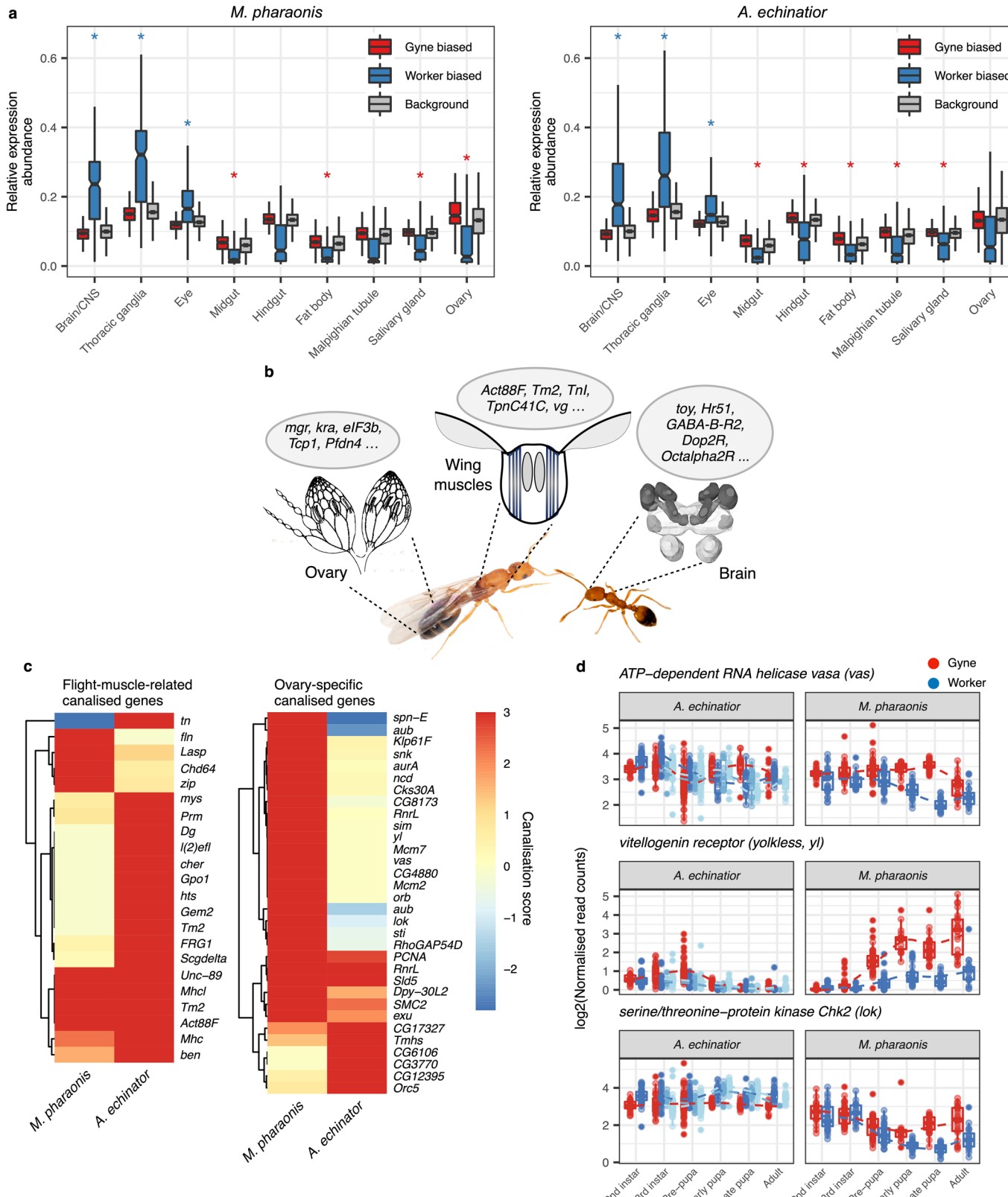

**Extended Data Fig. 6 | See next page for caption.**

**Extended Data Fig. 6 | Canalized genes play important roles in producing adaptive caste phenotypes. a**, Tissue-specific relative expression abundances (based on *Drosophila* gene orthologs) among canalized genes in *M. pharaonis* (left panel) and *A. echinatior* (right panel), plotted separately for gyne-biased and worker-biased genes (*M. pharaonis* gyne-biased, $n = 411$; *M. pharaonis* worker-biased, $n = 482$; *A. echinatior* gyne-biased, $n = 1119$, and *A. echinatior* worker-biased, $n = 899$). Compared to the whole-genome background (*M. pharaonis* background genes, $n = 8887$; *A. echinatior* background genes, $n = 7651$), worker-biased canalized genes had a significantly higher relative expression in the brain, eyes, and thoracic ganglia in both ant species (one-sided t-tests; $P_{M. pharaonis, brain} = 2.04 \times 10^{-93}$; $P_{M. pharaonis, thoracic ganglia} = 2.29 \times 10^{-87}$; $P_{M. pharaonis, eye} = 3.04 \times 10^{-31}$; $P_{A. echinatior, brain} = 9.20 \times 10^{-120}$; $P_{A. echinatior, thoracic ganglia} = 2.73 \times 10^{-181}$; $P_{A. echinatior, eye} = 1.74 \times 10^{-130}$). In *M. pharaonis*, gyne-biased canalized genes had a significantly higher relative transcript abundance in the midgut, fat body and ovaries (one-sided t-tests; $P_{M. pharaonis, midgut} = 7.01 \times 10^{-5}$; $P_{M. pharaonis, fat body} = 1.93 \times 10^{-6}$; $P_{M. pharaonis, ovary} = 3.04 \times 10^{-10}$). However, in *A. echinatior*, gyne-biased canalized genes showed no difference with background genes for their relative transcript abundance in ovaries (one-sided t-tests; $P = 0.97$), consistent with the workers having retained smaller ovaries. Box plots show the median (centre line), 25% and 75% quartiles (boxes) and outermost values (whiskers); $^*P < 0.01$, red for higher relative expression abundance in gyne-biased genes, blue for in worker-biased genes. **b**, Diagrammatic illustration

of gyne-biased canalized genes being associated with traits in ovaries and wing muscles, whereas worker-biased canalized genes are associated with brain function and behaviour (see Supplementary Table 4 for full list of canalized genes). **c**, Canalization scores for flight related (left) and ovary specific (right) genes in the two ant species. Flight related genes were identified based on their *D. melanogaster* homologues associated either with flight performance itself or with striated muscle functionality (the crucial wing muscles tissue in insects). Ovary specific genes were genes having > 30% expression abundance in ovaries of *D. melanogaster* females compared to the sum of their expression in all tissues (see **Methods**). Colours of cells represents the canalization score, ranging from −3 (blue, canalized in worker-biased direction) to 3 (red, canalized in gyne-biased direction). Canalization scores in *A. echinatior* were calculated by comparing gyne and small worker transcriptomes. **d**, Developmental expression dynamics of *ATP- dependent RNA helicase vasa* (*vas*), *vitellogenin receptor* (*yl*) and *serine/ threonine-protein kinase Chk2* (*lok*) in the two ant species. All three genes are ovary specific with high expression abundance in *D. melanogaster* ovaries. Although these three genes showed increasing gyne-biased canalization in *M. pharaonis* as development proceeds, there was little expression difference between gyne and worker individuals in *A. echinatior*, except for *yl* in prepupae and to a lesser extent also in 3$^{rd}$ instar larvae.

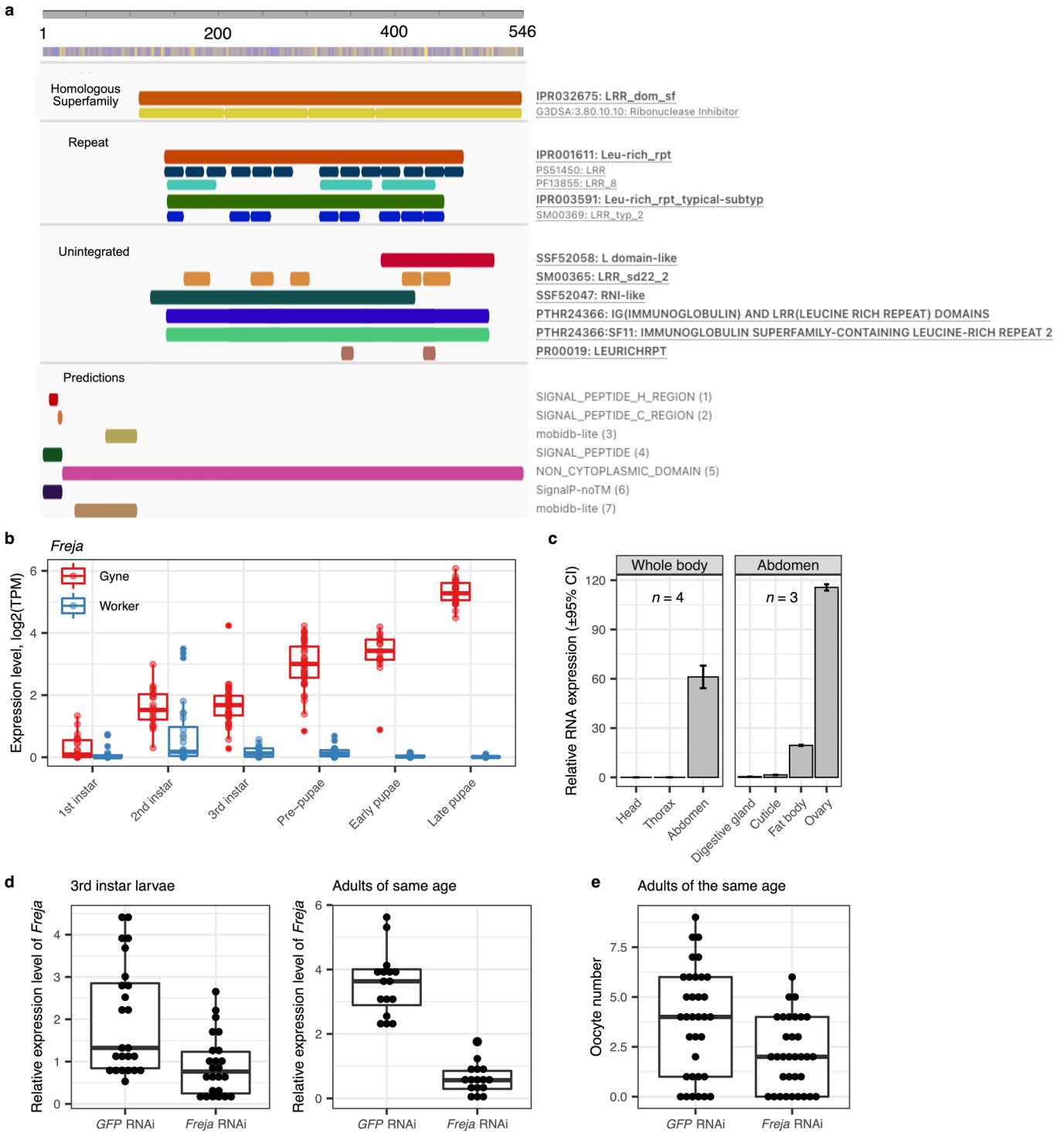

**Extended Data Fig. 7 | *Freja's* functional role in canalizing gyne phenotypes.**
**a**, Predicted functional domains of the protein encoded by *Freja* (LOC10587931), annotated with InterPro (see Methods). *Freja* contains a signal peptide domain at the N terminus, indicating secretion or membrane insertion. In addition, Freja contains a leucine-rich-repeat domain, suggesting its role in protein binding. **b**, *Freja* is the most strongly canalized gene in *M. pharaonis*, showing an increasing between-caste expression difference and a decreasing within-caste expression variance as development proceeds ($n_{gyne}$ = 168 and $n_{worker}$ = 161). The caste identities in 1st instar larvae are based on BPA prediction. **c**, Tissue-specific RT-PCR quantification of *Freja* transcript abundance in adult gynes, showing

*Freja's* expression is restricted to the abdomen and especially highly abundant in the ovaries. **d**, RT-PCR quantification of the efficiency of *Freja*-RNAi in 3rd instar larvae (left) and adult gynes (right). Compared to the *GFP*-RNAi control group, *Freja*-RNAi significantly reduced the expression level of *Freja* (p < 1e-3 in one-way ANOVAs in each age group). **e**, Compared with the control group, *Freja*-RNAi significantly reduced the number of yolky oocytes in adult gynes (p = 0.004 in one-way ANOVA). For Extended Data Figure 7b–e, box plots show the median (centre line), 25% and 75% quartiles (boxes), outermost values (whiskers) and data points (overlapping with box and whiskers).

**Extended Data Table 1 | Experimental design**

| Developmental stage | | Number of individual replicates | | | RNA extraction method | Average RNA concentration (ng/ul) | Average amount of RNA sequencing data (Gb per replicate) |
|---|---|---|---|---|---|---|---|
| | | Unknown caste | Gyne | Worker (Small/Medium/Large in *A. echinatior*) | | | |
| *M. pharaonis* | | | | | | | |
| Embryo | 0 – 12 h | 27 | | | | 0.1 | 8 |
| | 12 – 24 h | 21 | | | | 0.2 | 7 |
| | 36 – 48 h | 14 | | | | 0.2 | 6 |
| | 60 – 72 h | 21 | | | | 0.4 | 9 |
| | 84 – 96 h | 19 | | | PicoPure RNA kit | 0.2 | 9 |
| | 108 – 120 h | 14 | | | | 0.4 | 11 |
| | 132 – 144 h | 20 | | | | 0.4 | 5 |
| | 156 – 168 h | 19 | | | | 0.7 | 6 |
| | 180 – 192 h | 35 | | | | 0.4 | 8 |
| Larva | 1st instar | 53 | | | | 1.9 | 10 |
| | 2nd instar | | 49 | 26 | | 2.5 | 10 |
| | 3rd instar | | 37 | 17 | | 5.1 | 9 |
| | Pre-pupa | | 40 | 38 | RNeasy micro kit | 13.0 | 10 |
| Pupa | Early | | 20 | 29 | | 24.8 | 10 |
| | Late | | 30 | 25 | | 26.5 | 11 |
| Adult | Imago | | 24 | 16 | | 3.5 * | 9 |
| *A. echinatior* | | | | | | | |
| Larva | 1st instar | 40** | | | | 3.5 | 10 |
| | 2nd instar | 62** | | | | 23.5 | 9 |
| | 3rd instar | | 24 | 41/10/27 | | 123.5 | 7 |
| | 4th instar | | 26 | | RNeasy micro kit | 673.7 | 6 |
| | Pre-pupa | | 29 | 33/16/36 | | 358.3 | 7 |
| Pupa | Early | | 18 | 14/14/21 | | 380.1 | 12 |
| | Late | | 18 | 17/16/16 | | 261.8 | 11 |
| Adult | Imago | | 16 | 24/1/15 | | 294.2 | 11 |
| *D. melanogaster* | | | | | | | |
| | | Unknown sex | | Female | | | |
| Embryo | 1 h | 28 | | | | 2.7 | 7 |
| | 3 h | 24 | | | | 7.3 | 8 |
| | 6 h | 21 | | | | 4.4 | 7 |
| | 9 h | 18 | | | | 4.0 | 6 |
| | 12 h | 16 | | | PicoPure RNA kit | 3.6 | 7 |
| | 15 h | 18 | | | | 4.0 | 7 |
| | 18 h | 22 | | | | 3.4 | 7 |
| | 21 h | 21 | | | | 3.7 | 8 |
| Larva | 1st instar (24 – 44 h) | 79 | | | | 7 | 10 |
| | 2nd instar (50 – 68 h) | | | 21 | | 8.8 | 10 |
| | 3rd instar (Pre-pupa) (80 – 116 h) | | | 36 | RNeasy micro kit | 19.8 | 11 |
| Pupa | Early (128 h) | | | 18 | | 26.2 | 8 |
| | Late (140 h) | | | 14 | | 23.9 | 8 |
| Adult | Imago (180 h) | | | 14 | | 16.6 | 9 |

**Extended Data Table 2 | Determination of developmental stages and caste identities in the two ant species**

| Stage | | Caste | Head capsule width range | Body length range | Morphological characters |
|---|---|---|---|---|---|
| *M. pharaonis* | | | | | |
| Embryos | | U | | Not measured | |
| Larva | 1st instar | U | 0.11 – 0.15 mm | < 0.4 mm | |
| | 2nd instar | S | 0.16 – 0.23 mm | 0.5 – 1.2 mm | No body hairs |
| | | W | 0.15 – 0.20 mm | 0.4 – 0.7 mm | With body hairs |
| | 3rd instar | S | 0.23 – 0.32 mm | 1.2 – 3.1 mm | No body hairs |
| | | W | 0.20 – 0.28 mm | 0.7 – 2.1mm | With body hairs |
| | Pre-pupa | S | | 1.9 – 2.6 mm | Gut empty & No body hairs |
| | | W | | 1.3 – 1.7 mm | Gut empty & With body hairs |
| Pupa | Early | G | | 2.6 – 2.9 mm | White cuticle with gyne morphology |
| | | W | | 1.4 – 1.8 mm | White cuticle with worker morphology |
| | Late | G | | 2.6 – 2.9 mm | Dark cuticle with gyne morphology |
| | | W | | 1.4 – 1.8 mm | Dark cuticle with worker morphology |
| Adult | Imago | G | | Not measured | |
| | | W | | Not measured | |
| *A. echinatior* | | | | | |
| Larva | 1st instar | F | 0.18 – 0.31 mm | 0.5 – 0.9 mm | |
| | 2nd instar | F | 0.30 – 0.60 mm | 0.8 – 1.9 mm | |
| | 3rd instar | G | 0.31 – 0.57 mm | 2.0 – 5.0 mm | Curly hairs cover the whole body; Gyne type body shape (Abdomen is larger than the frontal body). |
| | | SW | 0.41 – 0.54 mm | 1.4 – 2.9 mm | No curly hairs; Few Y-shaped hairs on ventral thorax. No gyne type body shape. |
| | | MW | 0.46 – 0.51 mm | 3.1 – 3.9 mm | Same as above. |
| | | LW | 0.47 – 0.52 mm | 4.0 – 5.3 mm | Same as above. |
| | 4th instar | G | 0.49 – 0.56 mm | 5.0 – 7.2 mm | Curly hairs cover the whole body; Gut visible. |
| | Pre-pupa | G | | 6.3 – 7.2 mm | Curly hairs cover the whole body; Gyne type body shape. Developing legs and eye pigmentation are visible. Gut empty |
| | | SW | | 2.0 – 3.0 mm | Developing legs and eye pigmentation are visible. Gut empty |
| | | MW | | 3.0 – 4.0 mm | Same as above. |
| | | LW | | 4.0 – 5.2 mm | Same as above. |
| Pupa | Early | G | | 7.5 – 8.2 mm | |
| | | SW | | 2.1 – 3.0 mm | |
| | | MW | | 3.1 – 3.9 mm | |
| | | LW | | 4.1 – 6.1 mm | |
| | Late | G | | 7.3 – 8.1 mm | |
| | | SW | | 2.2 – 3.0 mm | |
| | | MW | | 3.1 – 4.0 mm | |
| | | LW | | 4.0 – 5.7 mm | |
| Adult | Imago | G | | 6.9 – 8.2 mm | |
| | | SW | | 2.0 – 3.9 mm | |
| | | MW | | 4.0 mm (only one sample) | |
| | | LW | | 4.1 – 7.0 mm | |

Jacobus Jan Boomsma
Guojie Zhang

# Reporting Summary

## Statistics

For all statistical analyses, confirm that the following items are present in the figure legend, table legend, main text, or Methods section.

| n/a | Confirmed | |
|---|---|---|
| ☐ | ☒ | The exact sample size (*n*) for each experimental group/condition, given as a discrete number and unit of measurement |
| ☐ | ☒ | A statement on whether measurements were taken from distinct samples or whether the same sample was measured repeatedly |
| ☐ | ☒ | The statistical test(s) used AND whether they are one- or two-sided *Only common tests should be described solely by name; describe more complex techniques in the Methods section.* |
| ☐ | ☒ | A description of all covariates tested |
| ☐ | ☒ | A description of any assumptions or corrections, such as tests of normality and adjustment for multiple comparisons |
| ☐ | ☒ | A full description of the statistical parameters including central tendency (e.g. means) or other basic estimates (e.g. regression coefficient) AND variation (e.g. standard deviation) or associated estimates of uncertainty (e.g. confidence intervals) |
| ☐ | ☒ | For null hypothesis testing, the test statistic (e.g. *F*, *t*, *r*) with confidence intervals, effect sizes, degrees of freedom and *P* value noted *Give P values as exact values whenever suitable.* |
| ☒ | ☐ | For Bayesian analysis, information on the choice of priors and Markov chain Monte Carlo settings |
| ☒ | ☐ | For hierarchical and complex designs, identification of the appropriate level for tests and full reporting of outcomes |
| ☒ | ☐ | Estimates of effect sizes (e.g. Cohen's *d*, Pearson's *r*), indicating how they were calculated |

*Our web collection on statistics for biologists contains articles on many of the points above.*

## Software and code

Policy information about availability of computer code

| Data collection | No software was used to collect data. |
|---|---|
| Data analysis | Custom code has been deposited in: https://github.com/BitaoQiu/devo-ants

The following software were used to analysis the data:

Morphological measurements: Adobe Photoshop CC 19.1.6 & ImageJ 1.53c
Imaging processing for fluorescence in situ hybridization: Fiji/ImageJ 1.53c
Microsatellite loci analysis: GeneMapper 4.0

Genome analyses:
Genome annotation: GeMoMa (ver. 1.7.1)
Ortholog detection: Orthofinder (ver. 2.5.4)
Sequence alignment: BLAST (ver. 2.12.0)
Identification of phylogenetic origin of genes:
Multiple sequence alignment: T-coffee (ver. 13.45.0)
Gene tree construction: IQ-TREE (ver. 2.1.4)

RNAseq analyses:
RNAseq reads quality control: SOAPnuke (ver. 2.0.7)
Transcriptome profiling: Salmon (ver. 1.4.0)
RNAseq read normalization, Variance stabilizing transformation, and DEG detection: DESeq2 (ver. 1.32.0)
Construction of developmental trajectory network from transcriptomes: igraph (ver. 1.2.9) |

Between-stage expression level normalization: sva (ComBat) (ver. 3.40.0)
Threshold regression model: chngpt (ver. 2021.5-12)
Robust linear regression: rlm from MASS (ver. 7.3)
Functional enrichment analysis: clusterProfiler (ver. 4.0.5)

For manuscripts utilizing custom algorithms or software that are central to the research but not yet described in published literature, software must be made available to editors and reviewers. We strongly encourage code deposition in a community repository (e.g. GitHub). See the Nature Portfolio guidelines for submitting code & software for further information.

## Data

Policy information about availability of data

All manuscripts must include a data availability statement. This statement should provide the following information, where applicable:
- Accession codes, unique identifiers, or web links for publicly available datasets
- A description of any restrictions on data availability
- For clinical datasets or third party data, please ensure that the statement adheres to our policy

RNAseq data that support the findings of this study have been deposited in GenBank with the BioProject accession codes PRJNA767561 (https://dataview.ncbi.nlm.nih.gov/object/PRJNA767561?reviewer=knbfs1f376d1idqf5crfn1ke5s)

# Field-specific reporting

Please select the one below that is the best fit for your research. If you are not sure, read the appropriate sections before making your selection.

☒ Life sciences        ☐ Behavioural & social sciences        ☐ Ecological, evolutionary & environmental sciences

For a reference copy of the document with all sections, see nature.com/documents/nr-reporting-summary-flat.pdf

# Life sciences study design

All studies must disclose on these points even when the disclosure is negative.

| Sample size | In total 1921 individual transcriptome samples, including: 819, 629, and 491 samples from M. pharaonis, A. echinatior and D. melanogaster, respectively, which secured having ca. 30 samples per stage per caste and a minimum of 30 samples per stage when caste phenotypes could not be determined. |
|---|---|
| Data exclusions | 63 samples were removed due to poor RNAseq quality (having within-stage Spearman correlation coefficients with other transcriptomes < 0.8).<br><br>Although RNA extraction, cDNA library construction and sequencing were all done with similar procedures (except for embryos and 1st instar samples for which we used a different extraction kit), we noticed a systematic expression difference for samples that had been sequenced before July 2018 (Batch A; 925 samples), before April 2019 (Batch B; 329 samples) and afterwards (Batch C;169 samples), producing three technical batches that could potentially confound the comparative analyses, especially among the A. echinatior samples.<br><br>We therefore excluded Batch B and C from cross-stage variation comparison while retained them for between-caste expression difference analysis, because the latter analysis can be adjusted for batch-effects by partial linear regression (see Methods for details). |
| Replication | We successfully verified our computational predictions of early caste marker genes with RNA fluorescence in situ hybridization.<br><br>We used quantitative reverse transcription PCR (RT-qPCR) to verify the top 10 canalized genes (genes with increasing between-caste expression divergence) and achieved matches in all cases.<br><br>For the roles of JH and Freja in caste canalization regulation, a minimum of four experimental replications have been conducted and all produced consistent results that confirmed our transcriptomic findings. |
| Randomization | Samples of each ant species were randomly collected from the same two (M. pharaonis, D03 and 4030) or three (A. echinatior, Ae150, Ae394 and Ae506) colonies.<br><br>Samples of the same developmental stages were randomized and processed with the same experimental procedures (RNA extraction, cDNA library construction and RNA sequencing), so that gyne and worker samples of the same stage were always randomized with minimal technical batch effect.<br><br>For the RNAi, JHA and precocene I experiments, experimental and control group individuals were randomly collected from the same colonies. During experiments, experimental and control group individuals were fed with the same food but reared separately, because we needed to add workers to take care of the larvae, which would have become mixed (between experimental and control groups) if they had been reared together. |
| Blinding | Blinding was not relevant to our study because we were examining the gyne-worker caste differentiation process, where caste identities need to be identified beforehand. For BPA (computational prediction of caste fate) and experimental validation of caste marker genes among early stage individuals, individuals' caste fates were blind to the experimenters. |

# Reporting for specific materials, systems and methods

We require information from authors about some types of materials, experimental systems and methods used in many studies. Here, indicate whether each material, system or method listed is relevant to your study. If you are not sure if a list item applies to your research, read the appropriate section before selecting a response.

## Materials & experimental systems

| n/a | Involved in the study |
|-----|----------------------|
| ☒ | ☐ Antibodies |
| ☒ | ☐ Eukaryotic cell lines |
| ☒ | ☐ Palaeontology and archaeology |
| ☐ | ☒ Animals and other organisms |
| ☒ | ☐ Human research participants |
| ☒ | ☐ Clinical data |
| ☒ | ☐ Dual use research of concern |

## Methods

| n/a | Involved in the study |
|-----|----------------------|
| ☒ | ☐ ChIP-seq |
| ☒ | ☐ Flow cytometry |
| ☒ | ☐ MRI-based neuroimaging |

## Animals and other organisms

Policy information about studies involving animals; ARRIVE guidelines recommended for reporting animal research

| | |
|---|---|
| Laboratory animals | We used Monomorium pharaonis from two colonies (D03 and 4030), both derived from interbreeding a global variety of genetic lineages in 2004, after which colonies have been kept in captivity at the University of Copenhagen at 27 °C and 50 % relative humidity throughout the experiments. Individual samples were collected between January 2017 and March 2018. Sample collection included all developmental stages (from 0-3 hour embryos to newly emerged adults) and gynes, workers and males of the experimental colonies. Caste identities of individuals were identified with morphological characters (for larvae, pupae, and adults), and sexes of individuals were identified with morphological characters (for pupae and adults) or microsatellite genotyping (for larvae) to determine ploidy.<br><br>We used Acromyrmex echinatior from three colonies (Ae150, Ae394 and Ae506), all collected in Gamboa, Panama between 2001 to 2011. Colonies were kept at 25 °C and 70 % relative humidity and were fed bramble leaves, rice and apples tree time a week. Individual samples were collected between March 2016 and November 2018. Sample collection included major developmental stages (from 1st instar larvae to newly emerged adults) and gynes, small/medium/large workers and males. Caste identities of individuals were identified with morphological characters and sexes of individuals were identified with morphological characters (for late stage larvae, pupae and adults) or microsatellite genotyping (for 1st and 2nd instar larvae).<br><br>We used Drosophila melanogaster with inbred wild-type genetic background Canton-S. Fly cultures were kept at 25 °C and 60% relative humidity throughout the experiment, with a 12-hour/12-hour light/dark cycle on standard Drosophila medium. Sample collection included all developmental stages (from 1 hour embryos to newly emerged adults) and both sexes of the experimental animals. Sexes of individuals were identified with morphological characters (for adults) or genotyping (for larvae and pupae). |
| Wild animals | *Provide details on animals observed in or captured in the field; report species, sex and age where possible. Describe how animals were caught and transported and what happened to captive animals after the study (if killed, explain why and describe method; if released, say where and when) OR state that the study did not involve wild animals.* |
| Field-collected samples | *For laboratory work with field-collected samples, describe all relevant parameters such as housing, maintenance, temperature, photoperiod and end-of-experiment protocol OR state that the study did not involve samples collected from the field.* |
| Ethics oversight | *Identify the organization(s) that approved or provided guidance on the study protocol, OR state that no ethical approval or guidance was required and explain why not.* |

Note that full information on the approval of the study protocol must also be provided in the manuscript.

