## [Peer Review File · Nature Ecology & Evolution]

Peer Review Information

Journal: Nature Ecology & Evolution

Manuscript Title: Canalized gene expression during development mediates caste differentiation in ants

Corresponding author name(s): Bitao Qiu, Jacobus J. Boomsma, Guojie Zhang

Editorial Notes:

Reviewer Comments & Decisions:

Decision Letter, initial version:

19th April 2022

Dear Guojie,

Your manuscript entitled "Canalized gene expression during development mediates caste differentiation in ants" has now been seen by three reviewers, whose comments are attached. The reviewers have raised a number of concerns which will need to be addressed before we can offer publication in Nature Ecology & Evolution. We will therefore need to see your responses to the criticisms raised and to some editorial concerns, along with a revised manuscript, before we can reach a final decision regarding publication.

We therefore invite you to revise your manuscript taking into account all reviewer and editor comments. Please highlight all changes in the manuscript text file in Microsoft Word format.

* Include a "Response to reviewers" document detailing, point-by-point, how you addressed each

reviewer comment. If no action was taken to address a point, you must provide a compelling argument. This response will be sent back to the reviewers along with the revised manuscript.

* If you have not done so already please begin to revise your manuscript so that it conforms to our Article format instructions at <http://www.nature.com/natecolevol/info/final-submission>. Refer also to any guidelines provided in this letter.

[REDACTED]

Nature Ecology & Evolution is committed to improving transparency in authorship. As part of our efforts in this direction, we are now requesting that all authors identified as 'corresponding author' on published papers create and link their Open Researcher and Contributor Identifier (ORCID) with their account on the Manuscript Tracking System (MTS), prior to acceptance. ORCID helps the scientific community achieve unambiguous attribution of all scholarly contributions. You can create and link your ORCID from the home page of the MTS by clicking on 'Modify my Springer Nature account'. For more information please visit www.springernature.com/orcid.

[REDACTED]

Reviewer expertise:

Reviewer #1: genomics of social insects, including cast differentiation

Reviewer #2: molecular basis of insect social evolution, comparative genomics

Reviewer #3: insect evo-devo, developmental genetics

Reviewers' comments:

Reviewer #1 (Remarks to the Author):

This work has put together an impressive number of developmental transcriptomes of ants to describe gene expression dynamics during caste differentiation. The main results show how caste expression profiles diverge during development, analogously to differentiation during development in multicellular development. This main finding is supported and complemented by mechanistic data from manipulations of development with hormone treatments and RNA-interference of key genes and visualization of their expression patterns. This work has the potential to be a landmark in sociogenomics, but there are several shortcomings that need attention first.

First, I have a worry about a potential confounding factor for one of the main results on canalization (e.g. Figure 3a): is it possible that quality of the libraries covaries with developmental stage? If (due to e.g. lower yield during extraction) there is more technical variation in the earlier brood stages (smaller individuals), this could explain at least a part of this observation. Please provide analyses that confirm this is not the case. The pattern shown seems clear so it would be very surprising if it would be completely driven by technical variation, but this should be checked. Please also provide information on how the numbers of discarded transcriptomes due to low quality were distributed across life stages and castes.

Second, many of the key results seem to be based on visual inspection of patterns (Especially figure 1, but also Figure 3 and Extended data figure) rather than statistical analyses – claims based on these figures need statistical support to be convincing, and going into more detailed comments on the results seems pointless before the statistical support for results and clear descriptions of the statistical tests are shown.

Third, I wonder whether there is an element of circularity in making claims on transcriptome similarities across species at any developmental stage, when the developmental stages are matched based on their similarity in the first place (lines 785 -> in the methods). Please explain whether or not this could be a problem. Similarly, in a study where the focus is so heavily on variation within and between sample classes, is it appropriate to exclude samples that do not correlate highly enough with other samples in their sample class. Please justify.

Finally, the logical jump from presented results to purifying selection being the evolutionary cause for the patterns is a quite long. This should be toned down or supported more strongly with analyses.

Reviewer #2 (Remarks to the Author):

3This work provides the most comprehensive dataset to date (to my knowledge) showing developmental stages in not one, but two social ant species. This is a really exciting development for the field and provides a lot of novel analysis that is clearly lacking from existing research, and sets the standard for future work.

Overall, the paper is well written and is exciting to follow throughout, providing multiple informative analyses in one paper that could be considered multiple papers combined.

My main criticisms come from some terminology used and the need for some additional information in methods, tables and figure, that with some refining/editing, would be a big improvement for the reader, and prevent any misunderstandings (most likely just on my part, a lot of the issues are probably addressed, but I could not find the information readily).

Main / general comments:

- Canalization is not defined in the introduction, with the definition only given on line 218 within the results section (see next point). I feel it would help the reader to explain what 'canalisation' means early on in the paper, so we can think about what we should expect. In addition, what is the null hypothesis? What would we expect to find in two castes that are becoming more different over developmental time?. Is it not expected that genes will become more differential as the castes become more physically and behaviorally different?
- The authors definition for canalization is provided as: 'the statistical tendency for individual transcriptomes to start with a unimodal (pluripotent) distribution and gradually change to a bimodal (phenotypically committed) distribution with increasingly distinct peaks as development proceeds'. I cannot find the source of this definition in the literature. Is this your own definition or is there a reference? Again, maybe if this was introduced explicitly, it would be clearer.
- I would have been interested to know if the key genes that are differential in the 'adult' workers and gynes, are changing at the early stages or not. I feel with this data you could have provided information about the way gene expression changes over time. e.g. using Mfuzz or similar gene expression time course software. As shown in this paper: <https://www.ncbi.nlm.nih.gov/pmc/articles/PMC6535812/>. Though I appreciate that this is an extra step and not necessary to justify your conclusions.
- I think the 'genome-wide developmental potential (Δ)' statistic needs to be better defined in the main text, as to how this has been calculated, as it is critical to understanding the figure 3. Line 223
- Line 317: How did you work out if a gene was hymenoptera specific. I couldn't find this in methods, but may have missed it.
- Line 302: Supp figure 6 does not really show that gyne canalized genes are specifically upregulated in ovary. This may not take anything away from your point, but gynes are essentially like the background for these categories, and it's the workers who vary from the background.

Figures

- Figure 1a. I really like the developmental trajectories, it is an interesting way to show this data, instead of "just another PCA". However, I feel the legend is not clear enough and I was initially confused by what is shown. There is no explanation for what the length of edges mean in legend or methods (e.g. do eggs have a greater distance to 12-24 hour embryos?, than 12-24h to 36-48h), I guess so, I assume the lengths are similar to the colour of the edges, is this correct?. This would suggest there is a massive transcriptomic transition from egg to 24hour stage, that is greater than the difference between adult castes, maybe representative of the genome activation stage (for example).
- Figure 1a. The legend should also include the comment from the methods, that the plot is of an 'undirected network', so the positioning of samples is somewhat random (e.g. eggs could be top right of plot).
- Figure 1b. I was really confused by PC1. Does this mean that you merged the two species gene expression matrices before running the PCA?. This seems a unusual to do it this way. It is not explained in the methods if this is so. If it is merged, then we need to know how many orthologs were used in this plot. Also, then how do you get two PC2 and two PC3 dimensions to plot in the figure, with one shared PC1. Maybe I am confused, if so, this needs better explanation.
- Figure 2 is great. Really clear.
- Figure 3a. Third instar larvae have an unusual pattern. There seem to be many BPA predicted workers, that have a body size similar to gynes and some with higher gyne potential. At the pre-pupal stage all of these large predicted workers have gone. Can you explain why we see this pattern, did these individuals die?, did they convert into gynes later on? Or do you think they were wrong predictions of their future caste.
- Extended Data Fig. 6.: Is there a reason not to show the same plot for *A. echinator*. Even if it does not show the same pattern, I think it is important to see if there is some similar trend (if not significant).
- Fig 4.: It was interesting to see that many worker canalized genes are named 'LOC...', does this suggest that a lot of the genes maybe species specific or have unknown function in *Drosophila*. It would have been interesting to know more about these genes.

Methods and supp tables:

- Are all the genomes and annotations used in this paper publically available. E.g. genome and annotation file for *Acromyrmex echinator*, say 'in house'. I cannot find the link to an archive in methods.
- What is 'low RNA quality (RIN < 4)', needs explanation. Also, it would be helpful to include the

5numbers of individuals filtered by quality and by similarity to its stage. It is a shame just to remove individuals with less than 0.8 similarity, these could have been interesting too, and could potentially show that some individuals are not canalized.

- In the main text, line 171, you suggest there is a difference between 1st instar caste phenotypes, is that what is shown in Supp table 1, the headers suggest this is a caste gyne/worker (maybe adults)?. Maybe there are missing tabs, with the differential expression between the other stages?
- Supplementary tables need a readme, either within the files or elsewhere (if not already). I am confused about the contents of each Supplementary table.
- E.g. on line 237 you refer the 65 conserved genes between the two species, yet in this table, we have the FDR pvalues (with tabs for each stage) in both species, so how did you derive 65 genes, and which ones are these?

I would accept with minor revisions.

Reviewer #3 (Remarks to the Author):

In this study, the authors compare the transcriptomes of developing workers and gynes and demonstrate that the transcriptomes diverge early in development. They identify components of the juvenile hormone signaling pathway as being distinctly expressed at different body sizes and demonstrate that they play a role in maintaining the distinct caste developmental trajectories. In addition, they identified a gene, which they name Freja, as a gyne-biased gene, which when knocked down produces intermediate phenotypes. Overall, the manuscript is easy to follow and the figures are clear. The findings are quite interesting and the discovery of Freja as a potential gyne-specific gene is a valuable contribution. I did have a few comments that I think the authors should address.

- The framework the authors use to discuss the development of ant castes is interesting: they discuss how caste differentiation in ants may be analogous to diverging cell lines depicted by Waddington's epigenetic landscape. Although interesting, I wondered if their finding can be applied to all ant colonies. As the authors note, "In this ant species, caste is known to be determined 'blastogenically' in early embryos, unlike most other ants where caste phenotype is determined during larval development". *M. pharaonis*, which they focus on in this study, seems to have undergone some unique canalization events that make the larval stages more robust to environmental cues. How is caste determined in *A. echinator*? I think caste determination of the two species they use might be something that they might want to discuss at the beginning rather than bury in the methods.

- Line 38: Please provide the names of the two species

- Fig. 3c: It looks like the graph for E93 is labeled Eip93F. To keep things consistent, I suggest using E93 for this.

- The JH-related genes appear to show heterochronic shifts (Fig. 3 and Extended Figure Fig. 5A). In

6other words, the drop in JH and activation of E93 gene expression is delayed in the gyne relative to the workers. These results indicate to me that rather than the JH signaling pathway acting as “a key regulator for caste canalization in ants” (line 263), the heterochronic shift in the timing of shutdown of JH is the critical switch between worker and gyne differentiation. This doesn’t negate the important of JH in specifying gyne or worker phenotypes, but I think the data nice demonstrate that heterochronic shifts of JH signaling play an important role in generating the distinct castes. Please consider this interpretation.

- The authors have identified a new gene that they call Freja and demonstrate that it is necessary for gyne development. They claim that this gene is a master regulator of queen phenotype, but I think the evidence is somewhat weak to make this claim. Firstly, the RNAi phenotypes generated are gyne-worker intermediates, not workers. If the gene was indeed a master regulator, the RNAi phenotype should be expected to be a worker. I realize that this may not be experimentally feasible, but as presented, I don't think one can make that claim. Secondly, to cement this claim, it might be helpful to examine some of the canalized genes they identified in Freja knockdown vs control gyne larvae. If the target genes all show the worker gene expression in Freja RNAi animals, it would provide additional evidence that this gene acts as a master regulator of queen development.

- Line 374 (and also Abstract): The authors propose that JH signaling “play a crucial role in regulating much of the canalization process of caste phenotypes via the control of, and feedback by, individual body size in the larval stages.” I am not sure the authors have presented evidence that there is feedback from body size. Although I tend to agree with the notion that the JH titers in most insects appear to be correlated with body size, the idea that there is some sort of feedback is not as well established.

- Line 326: Please correct the apostrophe

- The methods section is written in active voice. Was this intentional? Typically, methods section is written using passive voice

- Gene expression can vary dramatically within an instar. Based on the Extended Data Table 1 and 2, the exact timing of when samples collected for larval transcriptomes is not clear. It looks like the larvae were collected randomly within an instar whereas during embryonic development, and prepupal and pupal stages, the timing of collection is much more precise. I am wondering if that might lead over- or underestimation of transcriptome similarity between castes at different time points? If so, this might be worth mentioning in the discussion.

*****END*****

Author Rebuttal to Initial comments

7Response to reviewers:

Reviewer #1:

This work has put together an impressive number of developmental transcriptomes of ants to describe gene expression dynamics during caste differentiation. The main results show how caste expression profiles diverge during development, analogously to differentiation during development in multicellular development. This main finding is supported and complemented by mechanistic data from manipulations of development with hormone treatments and RNA-interference of key genes and visualization of their expression patterns. This work has the potential to be a landmark in sociogenomics, but there are several shortcomings that need attention first.

First, I have a worry about a potential confounding factor for one of the main results on canalization (e.g. Figure 3a): is it possible that quality of the libraries covaries with developmental stage? If (due to e.g. lower yield during extraction) there is more technical variation in the earlier brood stages (smaller individuals), this could explain at least a part of this observation. Please provide analyses that confirm this is not the case. The pattern shown seems clear so it would be very surprising if it would be completely driven by technical variation, but this should be checked. Please also provide information on how the numbers of discarded transcriptomes due to low quality were distributed across life stages and castes.

Response: RNA yields in larvae were both lower and more variable than in pupae. This is not surprising because (1) RNA yields are expected to be correlated with the body mass of samples, and (2) larvae are still actively growing and thus have higher body mass variation than pupae. However, RNA yield variation cannot technically explain our observed transcriptomic data. First, RNA-seq is a proportional measurement that quantifies relative gene expression levels, which should be independent of RNA yield during extraction. Second, although we found an association (Pearson correlation coefficient = 0.53; $p = 0.11$ in a two-sided test), the RNA yield variation did not match the variation in developmental potential in our data. For example, while there was a higher RNA yield variation in 2nd instar gynes than in 2nd instar workers, their developmental potential variation exhibited an opposite pattern (see Figure R1a below). Third, all else being equal, compared to RNA yield, RNA integrity number (RIN), a widely used statistic that assesses in vitro RNA degradation, is a better quantification of technical variation at RNA level, and we found no correlation between RIN variation and observed developmental potential variation (Pearson correlation coefficient = -0.02; $p = 0.95$ for two-sided test) (see Figure R1b below).

Regarding discarded transcriptomes. Because we only sequenced RNA samples with RIN > 4, all sequenced transcriptomes were of good RNA quality. Among the sequenced samples, 33 (out of 1971) samples were discarded based on their transcriptomic similarity with other samples that could possibly suggest a technical artifact. In *M. pharaonis*, 12 (out of 879) samples were discarded for the same

8reasons, including six embryonic samples, one 2nd instar, one 3rd instar and two adult workers, one 3rd instar gyne, and one 3rd instar male. In *A. echinator*, 9 (out of 599) samples were discarded, including one 2nd instar larva (of unknown caste), one early and two late-stage pupal workers, and one 3rd instar, three 4th instar and one prepupal gynes. In *Drosophila melanogaster*, 12 (out of 493) samples were discarded, including 11 embryonic and one 1st instar larval samples.

These samples were discarded only to make sure we excluded potential technically induced variation, e.g., artifacts due to RNA extraction, cDNA library construction, and sequencing. Because these discarded samples amounted to < 2%, we do not expect they will have impacted our conclusions. We have now provided this information in the Method section under *Sample quality control*.

We have also taken care of batch effects. 498 samples that were generated in different sequencing batches (identified by PCA of transcriptomes and further confirmed by examining the experimental metadata) were retained for differentially expressed gene analysis (after batch correction, see Method section under *Correction for batch effects*), but excluded from transcriptomic variation analyses to minimize the effect of batch variation. Therefore, we are confident that the observed pattern of transcriptomic variation that we report represent biological variation among samples with minimal effects of technical artifacts.

Figure R1. a. Absolute values of residual values of RNA yields (left) and absolute values of residual values of developmental potential (right, derived from Figure 3A in the manuscript) at each developmental stage, separately for gyne and worker caste individuals. b. Absolute values of residual values of RNA integrity number (RIN), a widely used measurement for assessing RNA quality, at each developmental stage, which showed no association with the absolute values of residual values of developmental potential. Because residual values for a variable of interest reflect the deviation from its statistical mean, a high level of residual values in absolute values indicates a high variation for the variable of interest.

Second, many of the key results seem to be based on visual inspection of patterns (Especially figure 1, but also Figure 3 and Extended data figure) rather than statistical analyses – claims based on these figures need statistical support to be convincing, and going into more detailed comments on the results seems pointless before the statistical support for results and clear descriptions of the statistical tests are shown.

Response: Our developmental trajectory network (Figure 1a) is based on the Fruchterman-Reingold layout, a force-directed drawing algorithm that visualizes the pair-wise correlation coefficient matrix among all individual transcriptomic samples. We now provide a higher-level summary of the same

data based on the mean values of within-group and between-group correlation coefficients (see Figure R2 below), which supports our conclusion from the trajectory network that caste differentiation is largely a continuous process but with increased caste differences in later stages. For example, the mean values of within-group correlation coefficients were always higher than the mean values of between-group correlation coefficients, which indicates higher similarities within groups. And the correlation coefficients (similarity) between gynes and workers in the 3rd larval instar was larger than in late pupae, indicating more substantial between-caste differences in pupae. In addition, the mean values of between-group correlation coefficients for embryos, larvae, pre-pupae + early pupae, and late pupae + adults clustered into four distinct groups, supporting our conclusion that stage-specific transcriptomes are discrete. We have now added this information in Extended Data Fig. 1b.

Figure R2, a higher-level summary of the data of Figure 1a presenting a between-stage transcriptomic similarity matrix, based on the mean values of within-group and between-group correlation coefficients. Correlation coefficients within the same stage are always higher than between stages, and transcriptomic similarities are clustered by adjacent developmental stages, supporting our conclusion of discrete stage-specific transcriptomes.

For Figure 1b, PC2 values of both ant species were significantly associated with developmental stages (all p values $< 1e-8$) and caste identities ($p < 5e-4$ for both species) and there was a significant interaction

11

effect between developmental stage and caste identity on PC2 values ($p < 0.05$ for both species; two-way ANOVA). PC3 values in both species were also significantly associated with developmental stages (all p values $< 5e-3$; one-way ANOVA). We have now added this information in the main text.

For Figure 1c, the between-species transcriptomic similarities for gynes were significantly higher than for workers ($p < 0.01$ for all examined developmental stages; two-sided t-tests) and there was a significant difference for the between-species transcriptomic similarities across developmental stages ($p < 1e-7$; two-way ANOVA). We have now added this information in the main text.

For Figure 3a, transcriptomic developmental potential in gynes and workers exhibited significant divergence (difference) ($p < 0.05$; two-sided t-tests) from as early as 1st instar larvae onwards. This divergence increased across developmental stages (t -score increased from 3.7 in 1st larval instar to 45 in late pupa) and was statistically significant ($p < 1e-16$; two-way ANOVA) as overall trend. The variances of developmental potential in workers were always significantly higher than in gynes for all stages ($p < 0.05$; two-sided F-test). We have now added this information in the main text.

Third, I wonder whether there is an element of circularity in making claims on transcriptome similarities across species at any developmental stage, when the developmental stages are matched based on their similarity in the first place (lines 785 -> in the methods). Please explain whether or not this could be a problem. Similarly, in a study where the focus is so heavily on variation within and between sample classes, is it appropriate to exclude samples that do not correlate highly enough with other samples in their samples class. Please justify.

Response: While we matched developmental stages between species based on their overall transcriptomic similarities, we found that the best matched stages were always within the same category, e.g., larvae or pupae (see Figure R3a below), indicating an overall developmental similarity among the two examined ant species. Furthermore, our main findings of the between-species similarity analyses revealed that: (1) transcriptomic similarities in gynes were higher than in workers, and (2) transcriptomic similarities in larval stages were higher than in pupal stages. Because alignments in each developmental stage and each caste were done independently, the between-species similarity across different stages and different castes must be comparable. We observed a similar pattern for the same analysis without alignment (see Figure R3b below), showing that our findings are independent of our developmental stage alignments.

Regarding the influence of excluding samples that may have been technical artifacts, as we mentioned above, we excluded only 33 (out of 1971) samples that showed transcriptomic derivations that could have arisen for technical reasons. Regardless, we obtained very similar patterns with and without sample filtering for both transcriptomic similarity and developmental potential (see Figure R3c below), supporting the robustness of our findings.

Figure R3. a. Between-species pair-wise developmental transcriptomic similarity matrix in gynes (left panel) and workers (right panel). Colour of cells are based on their transcriptomic correlation coefficient (similarity) (red: high similarity; blue: low similarity). Both matrices show that transcriptomes of the same stage are similar to each other both within and between the two ant species (e.g., 3rd instar larvae in *M. pharaonis* are highly similar to 3rd and 4th instar larvae in *A. echinator*). b. Between-species transcriptomic similarity across different developmental stages, without cross species stage matching, showing a similar pattern as in our Figure 1c of the main manuscript. c. Developmental potential in *M. pharaonis*, using all samples (without sample filtering), showing that the pattern in Figure 3a (main manuscript) is robust to sample filtering.

Finally, the logical jump from presented results to purifying selection being the evolutionary cause for the patterns is a quite long. This should be toned down or supported more strongly with analyses.

Response: We have revised the sentence to stay closer to the phenomena that we directly studied. See Line 419 - 421.Reviewer #2:

This work provides the most comprehensive dataset to date (to my knowledge) showing developmental stages in not one, but two social ant species. This is a really exciting development for the field and provides a lot of novel analysis that is clearly lacking from existing research, and sets the standard for future work.

Overall, the paper is well written and is exciting to follow throughout, providing multiple informative analyses in one paper that could be considered multiple papers combined.

My main criticisms come from some terminology used and the need for some additional information in methods, tables and figure, that with some refining/editing, would be a big improvement for the reader, and prevent any misunderstandings (most likely just on my part, a lot of the issues are probably addressed, but I could not find the information readily).

Main / general comments:

- Canalization is not defined in the introduction, with the definition only given on line 218 within the results section (see next point). I feel it would help the reader to explain what ‘canalisation’ means early on in the paper, so we can think about what we should expect. In addition, what is the null hypothesis? What would we expect to find in two castes that are becoming more different over developmental time?. Is it not expected that genes will become more differential as the castes become more physically and behaviorally different?

Response: Thank you for the suggestion. We have now added the definition of canalization in the introduction section. According to Waddington’s landscape metaphor, caste differentiation is expected to proceed as a developmental bifurcation process, which leads to predictions that (1) the number of differentially expressed genes increases towards the later developmental stages, and (2) key genes for establishing caste phenotypes should exhibit less variation within each of the castes. Our study evaluates the degrees to which these two patterns are upheld so that inferring developmental canalization in association with caste differentiation is the logical conclusion. We also expected that gynes, the germline analogues of a colony, should be more canalized because they are essential for direct genetic inheritance of traits under natural selection. We predict that these phenomena should only occur in super-organismal social insects such as ants, yellowjacket wasps, corbiculate bees (except basal orchid bees) and higher termites, but not in non-superorganismal social insects that merely form societies with phenotypically plastic helper/breeder role differentiation. We have now made these hypotheses explicit in the

15introduction section (see Line 76 – 82 and Line 111 - 115).

- The authors definition for canalization is provided as: ‘the statistical tendency for individual transcriptomes to start with a unimodal (pluripotent) distribution and gradually change to a bimodal (phenotypically committed) distribution with increasingly distinct peaks as development proceeds’. I cannot find the source of this definition in the literature. Is this your own definition or is there a reference? Again, maybe if this was introduced explicitly, it would be clearer.

Response: This definition is our own. It is based on a classic organismal interpretation of canalization in metazoan bodies that we extrapolated to the higher superorganismal level, following hypotheses by William Morton Wheeler more than a century ago, but expressed in Waddington language. Waddington’s logic was that metazoan organismal differentiation starting from a zygote leads to the formation of discrete tissues regardless of minor environmental and genotypic disturbances, because natural selection has canalized these key developmental processes. Analogously, we hypothesized that higher-level developmental canalization will robustly secure the discrete gyne and worker phenotypes of ants, and that this process should be detectable by increasingly bimodal distributions of individual transcriptomes as individuals develop towards their adult phenotypes. We have now introduced this more explicitly (see Line 76 – 82 and Line 237 – 241).

- I would have been interested to know if the key genes that are differential in the ‘adult’ workers and gynes, are changing at the early stages or not. I feel with this data you could have provided information about the way gene expression changes over time. e.g. using Mfuzz or similar gene expression time course software. As shown in this paper: <https://www.ncbi.nlm.nih.gov/pmc/articles/PMC6535812/>. Though I appreciate that this is an extra step and not necessary to justify your conclusions.

Response: The gene expression differences between gynes and workers at the early stages can be found in Supplementary Table 4, where we have provided the stage-specific ratios of between-caste difference and within-caste variance for all the canalized genes. Some of the canalized genes, e.g., *Freja* and *SMYD3*, were differentially expressed as early as the 2nd larval instar. We have now provided a readme for each Supplementary table to better guide readers to make full use of our data.

- I think the ‘genome-wide developmental potential (\square)’ statistic needs to be better defined in the main text, as to how this has been calculated, as it is critical to understanding the figure 3. Line 223

Response: Thank you for pointing this out, we have now provided the definition and the calculation of developmental potential in the main text (see the legend of Fig. 3a).

• Line 317: How did you work out if a gene was hymenoptera specific. I couldn't find this in methods, but may have missed it.

Response: We identified the phylogenetic origins of genes by identifying their ortholog groups (the sets of genes including both orthologs and paralogs) descended from a single gene in the last common ancestor. Because all gene members of the *Freja* ortholog group were found exclusively within the Hymenoptera, we inferred that *Freja* is a hymenopteran order-specific gene. This information is provided in the Method section under *Identifying the phylogenetic origin of Freja*.

• Line 302: Supp figure 6 does not really show that gyne canalized genes are specifically upregulated in ovary. This may not take anything away from your point, but gynes are essentially like the background for these categories, and it's the workers who vary from the background.

Response: The reviewer is correct that worker-biased canalized genes have a higher deviation from the background genes, which is consistent with workers being the evolutionarily derived caste. However, the differences between gyne-biased canalised genes and background genes were significant ($p < 0.05$; two-sided t-tests). In Extended Data Fig. 6, we used red asterisks above each boxplot to indicate where gyne-biased canalized genes were significantly more highly expressed in the target tissue compared to the background genes.

Regarding to caste-differences for ovarian expressed genes, although the difference between gyne-biased genes and background genes were not as substantial as those between worker-biased genes and background genes, the gyne-biased canalised genes did have a significantly higher relative expression abundance ($p < 5e-8$; two-sided t-test; comparing mean relative expression abundance for gyne-biased canalised genes, worker-biased canalized genes, and background genes = 16.4%, 6.4%, and 13.2%, respectively).

Figures

• Figure 1a. I really like the developmental trajectories, it is an interesting way to show this data, instead

17of “just another PCA”. However, I feel the legend is not clear enough and I was initially confused by what is shown. There is no explanation for what the length of edges mean in legend or methods (e.g. do eggs have a greater distance to 12-24 hour embryos?, than 12-24h to 36-48h), I guess so, I assume the lengths are similar to the colour of the edges, is this correct?. This would suggest there is a massive transcriptomic transition from egg to 24hour stage, that is greater than the difference between adult castes, maybe representative of the genome activation stage (for example).

Response: Thank you for the comment. The plotted developmental trajectories are visualizing the pairwise correlation coefficients among all individual transcriptomes, and the colour of edges is based on the correlation coefficient between samples. Because the network layout is calculated with Fruchterman-Reingold layout, a force-directed drawing algorithm, samples (nodes) with high transcriptomic similarities are clustering together. Therefore, the reviewer is correct that the lengths of edges from eggs to 24-hour embryos is larger than the differences found at later stages. We have now added more explanation of these differences in the legend.

- Figure 1a. The legend should also include the comment from the methods, that the plot is of an ‘undirected network’, so the positioning of samples is somewhat random (e.g. eggs could be top right of plot).

Response: Yes, we have now added this information in the legend.

- Figure 1b. I was really confused by PC1. Does this mean that you merged the two species gene expression matrices before running the PCA?. This seems a unusual to do it this way. It is not explained in the methods if this is so. If it is merged, then we need to know how many orthologs were used in this plot. Also, then how do you get two PC2 and two PC3 dimensions to plot in the figure, with one shared PC1. Maybe I am confused, if so, this needs better explanation.

Response: Sorry for the confusion. The PCA is based on all individual transcriptomes (excluding embryos, because we did not collect embryos of *A. echinator*) of the two ant species (number of samples = 979), using all one-to-one orthologous genes (number of genes = 7838) of both species. Our method for orthologous gene detection can be found in the method section under the subheading *Detection of orthologs and homologs across species*.

PC1 of the PCA is driven by the transcriptomic difference between the two ant species and presented in the density plot of Figure 1B. PC2 and PC3 of the PCA were driven by the developmental differences and caste differences that were shared between the two ant species. We plotted these PC2 and PC3 values (from the same PCA run) separately for each ant species to illustrate the similarity between the two ant species in developmental and caste differentiation transcriptomes. We believe that our approach is essential for any cross-species comparison, also because a similar approach has been applied in comparative analyses of developmental transcriptomes across organs for a series of mammalian species (<https://doi.org/10.1038/s41586-019-1338-5>, Figure 1b). We have now provided more explanation in the legend of Figure 1b.

- Figure 2 is great. Really clear.

Response: Thank you!

- Figure 3a. Third instar larvae have an unusual pattern. There seem to be many BPA predicted workers, that have a body size similar to gynes and some with higher gyne potential. At the pre-pupal stage all of these large predicted workers have gone. Can you explain why we see this pattern, did these individuals die?, did they convert into gynes later on? Or do you think they were wrong predictions of their future caste.

Response: Caste identities of the third instar larvae were not based on BPA but on their known morphological characters (Extended Data Table 2) - we just confirmed samples' caste identities with our BPA prediction (Extended Figure 2C). We hypothesize that the overlapping developmental potential for workers and gynes in the larval stages is due to the relatively smaller number of DEGs at these early stages of development (only a few hundred DEGs), while developmental potential was calculated from full transcriptomes, which had high similarity at these early stages as well. We obtained a better separation when we only used caste differentially expressed genes at each stage but preferred to use whole transcriptomes to keep the different stages comparable and to show that developmental potential at later stages is much more canalized than at earlier stages of development. We have now made this clearer in the legend of Figure 3a.

- Extended Data Fig. 6.: Is there a reason not to show the same plot for *A. echinator*. Even if it does not show the same pattern, I think it is important to see if there is some similar trend (if not significant).

Response: Thank you for the suggestion. We have now included the figure for *A. echinator* (Extended Data Fig. 6a), which shows a similar pattern to *M. pharaonis* except that, compared with background genes, gyne-biased canalized genes in *A. echinator* are not significantly more highly expressed in ovaries. This is consistent with ovarian genes being more developmentally canalized in gynes of *M. pharaonis* than in gynes of *A. echinator*, which in turn relates to workers of the latter species having retained ovaries and the ability to lay unfertilized eggs.

• Fig 4.: It was interesting to see that many worker canalized genes are named ‘LOC...’, does this suggest that a lot of the genes maybe species specific or have unknown function in drosophila. It would have been interesting to know more about these genes.

Response: Yes, this is correct. Many of these worker-biased canalized genes lack homologous genes in *Drosophila* and their functions are thus unknown. This is consistent with our previous finding that worker-biased genes (in brains) are more evolutionary novel and more lineage specific (see Qiu et. al NEE, 2018).

Methods and supp tables:

• Are all the genomes and annotations used in this paper publically available. E.g. genome and annotation file for Acromyrmex echinator, say ‘in house’. I cannot find the link to an archive in methods.

Response: All genomes except that of *A. echinator* are available in NCBI. The genome and annotation files of *A. echinator* were still under preparation when we submitted the first version of our manuscript. We have now uploaded these genome and annotation files of *A. echinator* to CNGB Nucleotide Sequence Archive (CNSA: <https://db.cngb.org/cnsa>; accession number CNP0003055)

• What is ‘low RNA quality (RIN < 4)’, needs explanation. Also, it would be helpful to include the numbers of individuals filtered by quality and by similarity to its stage. It is a shame just to remove individuals with less than 0.8 similarity, these could have been interesting too, and could potentially show that some individuals are not canalized.

Response: RIN means RNA integrity number, a widely used statistic to assess in vitro RNA degradation (<https://doi.org/10.1186/1471-2199-7-3>), with a low RIN indicating degraded RNA. Our in-house

20experience showed that a threshold criterion of $RIN > 4$ provides a good balance between allowing large scale sampling to proceed while securing sufficient RNA quality for individual RNA samples. As we wrote above in response to Reviewer 1, we have in total filtered out only 33 (out of 1971) samples based on unexplainable within-stage transcriptomic similarity. Although we appreciate that these outlier samples might be inter-caste phenotypes that may provide interesting additional insight in the caste differentiation process, these outliers were so rare that we felt it was more important to remove potentially artifactual noise. We have uploaded the transcriptomes for these samples together with all other samples, and we have now provided the number of filtered-out samples at each stage in the Method section under *Sample quality control*.

- In the main text, line 171, you suggest there is a difference between 1st instar caste phenotypes, is that what is shown in Supp table 1, the headers suggest this is a caste gyne/worker (maybe adults)?. Maybe there are missing tabs, with the differential expression between the other stages?

Response: Supplementary Table 1 contained only the predicted caste differentially expressed genes for 1st instar larvae. Caste differentially expressed genes in other stages can be found in Supplementary Table 2. We have now provided a readme for each supplementary table to facilitate finding the source data.

- Supplementary tables need a readme, either within the files or elsewhere (if not already). I am confused about the contents of each Supplementary table. E.g. on line 237 you refer the 65 conserved genes between the two species, yet in this table, we have the FDR pvalues (with tabs for each stage) in both species, so how did you derive 65 genes, and which ones are these?

Response: Sorry for the confusion, the 65 conserved genes can be found in Supplementary Table 2, under the tab *Larval (body size controlled)*. In this tag, we reported all candidate genes ($n = 147$) with conserved caste expression differences across the two ant species (absolute value of \log_2 fold change between castes > 0.8 for both ant species; without filtering by FDR). While filtering by FDR can provide more robust results, i.e., the 65 genes that we reported in the main text, it might also lead to false negatives (see Supplementary Information of Qiu, et al. NEE, 2018, where we discussed the potential problem of using overlapping approaches for detecting conserved DEGs). We therefore reported all potential candidate genes in the supplementary file so readers interested in these details can get access to the alternative data sets. We have now provided readmes for all Supplementary tables.

I would accept with minor revisions.Reviewer #3:

In this study, the authors compare the transcriptomes of developing workers and gynes and demonstrate that the transcriptomes diverge early in development. They identify components of the juvenile hormone signaling pathway as being distinctly expressed at different body sizes and demonstrate that they play a role in maintaining the distinct caste developmental trajectories. In addition, they identified a gene, which they name Freja, as a gyne-biased gene, which when knocked down produces intermediate phenotypes. Overall, the manuscript is easy to follow and the figures are clear. The findings are quite interesting and the discovery of Freja as a potential gyne-specific gene is a valuable contribution. I did have a few comments that I think the authors should address.

- The framework the authors use to discuss the development of ant castes is interesting: they discuss how caste differentiation in ants may be analogous to diverging cell lines depicted by Waddington's epigenetic landscape. Although interesting, I wondered if their finding can be applied to all ant colonies. As the authors note, "In this ant species, caste is known to be determined 'blastogenically' in early embryos, unlike most other ants where caste phenotype is determined during larval development". *M. pharaonis*, which they focus on in this study, seems to have undergone some unique canalization events that make the larval stages more robust to environmental cues. How is caste determined in *A. echinator*? I think caste determination of the two species they use might be something that they might want to discuss at the beginning rather than bury in the methods.

Response: Thank you for this suggestion. While we indeed know that caste determination in *M. pharaonis* happens early in the embryonic stage (i.e. in the egg stage), and during the early larval stages for *A. echinator* (as our BPA showed, no later than the 2nd instar), the detailed mechanisms underlying caste determination in these two ant species are still unknown. We agree with the reviewer that *A. echinator* may be more representative for 'ants in general' than *M. pharaonis*, but also maintain that the major evolutionary transition to superorganismality in the common ancestor of all ants should have left clear signatures of caste canalization in most ants, except for a few derived lineages that secondary lost the queen-caste germline analogue (see also Qiu et al, NEE 2018).

This in contrast with lineages such as lower termites and paper wasps where caste developmental trajectories have remained plastic, so individuals are not committed to a specific caste phenotype for life. By using our methods, we have revealed candidate genes that are key to the caste differentiation process across the two (Myrmicinae subfamily) ant species that we studied. However, due to the challenges of collecting early-stage sexual larvae in *A. echinator* (which are embedded in fungus garden material) it

23remained difficult to establish genetic manipulation experiments in this species. Therefore, we have focused on the *Monomorium* for these functional experiments. We have now introduced the key differences between the two ant species in the introduction as requested. See Line 105 – 115.

- Line 38: Please provide the names of the two species

Response: We have now provided this information.

- Fig. 3c: It looks like the graph for E93 is labeled Eip93F. To keep things consistent, I suggest using E93 for this.

Response: We have now changed the label.

- The JH-related genes appear to show heterochronic shifts (Fig. 3 and Extended Figure Fig. 5A). In other words, the drop in JH and activation of E93 gene expression is delayed in the gyne relative to the workers. These results indicate to me that rather than the JH signaling pathway acting as “a key regulator for caste canalization in ants” (line 263), the heterochronic shift in the timing of shutdown of JH is the critical switch between worker and gyne differentiation. This doesn't negate the important of JH in specifying gyne or worker phenotypes, but I think the data nice demonstrate that heterochronic shifts of JH signaling play an important role in generating the distinct castes. Please consider this interpretation.

Response: We agree with the reviewer that our results indicate a heterochronic shift in the timing of JH shutdown and we have added this interpretation in the main text. See Line 280 – 282.

- The authors have identified a new gene that they call Freja and demonstrate that it is necessary for gyne development. They claim that this gene is a master regulator of queen phenotype, but I think the evidence is somewhat weak to make this claim. Firstly, the RNAi phenotypes generated are gyne-worker intermediates, not workers. If the gene was indeed a master regulator, the RNAi phenotype should be expected to be a worker. I realize that this may not be experimentally feasible, but as presented, I don't think one can make that claim. Secondly, to cement this claim, it might be helpful to examine some of the canalized genes they identified in Freja knockdown vs control gyne larvae. If the target genes all show the worker gene expression in Freja RNAi animals, it would provide additional evidence that this gene acts as a master regulator of queen development.

Response: Turning a gyne-destined larva into a full worker via RNAi will be experimentally impossible because phenotypic characters of gynes had already partially developed by the time of the RNAi experiment. Although our experimental results support the role of *Freja* in ovary functionality of *Monomorium* gynes/queens, we agree with the reviewer that the definition of ‘master regulator of queen phenotype’ is arguable. We have now toned down this term and changed it to *a crucial regulator*.

We agree with the reviewer that it is worth to reveal the target genes that *Freja* regulates and the functional mechanisms by which this happens. However, to clearly demonstrate the functional mechanisms of *Freja* would require an entire array of new follow-up experiments including RNA-seq after the RNAi experiment included in our present paper, as the reviewer suggests. This is out of the scope of our current manuscript, but we agree it can be our next step effort.

- Line 374 (and also Abstract): The authors propose that JH signaling “play a crucial role in regulating much of the canalization process of caste phenotypes via the control of, and feedback by, individual body size in the larval stages.” I am not sure the authors have presented evidence that there is feedback from body size. Although I tend to agree with the notion that the JH titers in most insects appear to be correlated with body size, the idea that there is some sort of feedback is not as well established.

Response: Sorry for the confusion. What we meant here is that body size at the prepupal stage (when individuals’ body size stops increasing) is the result of caste differentiation, but not the sole determinant for caste phenotypes as a recent opinion paper proposed (<https://doi.org/10.1016/j.tree.2020.11.010>). Our data show that caste differentiation occurs early in larval development and that gynes and workers, in spite of having similar body sizes in the early larval stages, already exhibited distinct phenotypic characters at both transcriptomic and anatomical levels that were independent of body size. We have now stated our view more explicitly. See Line 399- 401.

- Line 326: Please correct the apostrophe

Response: We have corrected this.

- The methods section is written in active voice. Was this intentional? Typically, methods section is written using passive voice

Response: We have now rewritten the method section in passive voice.

- Gene expression can vary dramatically within an instar. Based on the Extended Data Table 1 and 2, the exact timing of when samples collected for larval transcriptomes is not clear. It looks like the larvae were collected randomly within an instar whereas during embryonic development, and prepupal and pupal stages, the timing of collection is much more precise. I am wondering if that might lead over- or underestimation of transcriptome similarity between castes at different time points? If so, this might be worth mentioning in the discussion.

Response: Larval samples within the same instar were collected randomly to cover all developmental time points within each instar stage, where we have measured body length and head capsule width for all larval individuals, we were therefore able to use generalised linear models to identify caste differentially expressed genes adjusted for body-size.

Because we sampled gynes and workers within the same stage with the same procedure, the variation in gynes and workers are comparable within these stages. Furthermore, because the estimation of between-caste similarity is based on the average transcriptomic differences between castes, between-caste similarity should be comparable across stages as well. Having said that, the reviewer is correct in pointing out that our data would have been more accurate with a more fine-grained sampling schedule across age classes, but this was not feasible. We maintain, however, that our sampling and size-correction procedures have minimized the bias in our within versus across caste estimates of transcriptome variation for the different developmental stages.

Decision Letter, first revision:

24th June 2022

Dear Guojie,

Thank you for submitting your revised manuscript "Canalized gene expression during development mediates caste differentiation in ants" (NATECOLEVOL-220315998A). It has now been seen again by the original reviewers and their comments are below. The reviewers find that the paper has improved

26in revision, and therefore we'll be happy in principle to publish it in Nature Ecology & Evolution, pending minor revisions to satisfy the reviewers' final requests and to comply with our editorial and formatting guidelines.

[REDACTED]

Reviewer #1 (Remarks to the Author):

The authors have done a thorough job in responding to my concerns. I have two comments remaining.

1. The analyses concerning the confounding effect of technical variation are convincing. However, I think these analyses should be added to the manuscript as well - the problem of biological and technical variation being potentially confounded is a real issue in studies interested in transcriptome variability, and given the high expected impact of this paper it would be important to a) make people aware of this issue b) clearly show that it is unlikely to be a worry here

2. Higher levels of canalization in sexual vs. workers is an interesting finding and could benefit from more discussion. Especially, are the analogous findings from lower level organisms and their germline vs somatic cells?

Reviewer #2 (Remarks to the Author):

I am satisfied with the revisions. The authors have addressed my concerns about terminology used and added READMEs and additional text in figure legends that help the reader understand the data.

Reviewer #3 (Remarks to the Author):

The authors have addressed all of my comments in a satisfactory manner. Please note the following minor edits:

Line 304: "identifies" should be "identities"

Line 746: abbreviate "Drosophila melanogaster" as "D. melanogaster" since it's already been mentioned before?

Our ref: NATECOLEVOL-220315998A

30th June 2022

Dear Dr. Zhang,

Thank you for your patience as we've prepared the guidelines for final submission of your Nature Ecology & Evolution manuscript, "Canalized gene expression during development mediates caste differentiation in ants" (NATECOLEVOL-220315998A). Please carefully follow the step-by-step instructions provided in the attached file, and add a response in each row of the table to indicate the changes that you have made. Please also check and comment on any additional marked-up edits we have proposed within the text. Ensuring that each point is addressed will help to ensure that your revised manuscript can be swiftly handed over to our production team.

****We would like to start working on your revised paper, with all of the requested files and forms, as soon as possible (preferably within two weeks). Please get in contact with us immediately if you anticipate it taking more than two weeks to submit these revised files.****

In recognition of the time and expertise our reviewers provide to Nature Ecology & Evolution's editorial process, we would like to formally acknowledge their contribution to the external peer review of your manuscript entitled "Canalized gene expression during development mediates caste differentiation in ants". For those reviewers who give their assent, we will be publishing their names alongside the published article.

Nature Ecology & Evolution offers a Transparent Peer Review option for new original research manuscripts submitted after December 1st, 2019. As part of this initiative, we encourage our authors to support increased transparency into the peer review process by agreeing to have the reviewer comments, author rebuttal letters, and editorial decision letters published as a Supplementary item. When you submit your final files please clearly state in your cover letter whether or not you would like to participate in this initiative. Please note that failure to state your preference will result in delays in

28accepting your manuscript for publication.

Cover suggestions

As you prepare your final files we encourage you to consider whether you have any images or illustrations that may be appropriate for use on the cover of Nature Ecology & Evolution.

Nature Ecology & Evolution has now transitioned to a unified Rights Collection system which will allow our Author Services team to quickly and easily collect the rights and permissions required to publish your work. Approximately 10 days after your paper is formally accepted, you will receive an email in providing you with a link to complete the grant of rights. If your paper is eligible for Open Access, our Author Services team will also be in touch regarding any additional information that may be required to arrange payment for your article.

Please note that *Nature Ecology & Evolution* is a Transformative Journal (TJ). Authors may publish their research with us through the traditional subscription access route or make their paper immediately open access through payment of an article-processing charge (APC). Authors will not be required to make a final decision about access to their article until it has been accepted. [Find out more about Transformative Journals](https://www.springernature.com/gp/open-research/transformative-journals)

Authors may need to take specific actions to achieve [compliance with funder and institutional open access mandates](https://www.springernature.com/gp/open-research/funding/policy-compliance-faqs). If your research is supported by a funder that requires immediate open access (e.g. according to [Plan S principles](https://www.springernature.com/gp/open-research/plan-s-compliance)) then you should select the gold OA route, and we will direct you to the compliant route where possible. For authors selecting the subscription publication route, the journal's standard licensing terms will need to be accepted, including [self-archiving-and-license-to-publish](https://www.nature.com/nature-portfolio/editorial-policies/self-archiving-and-license-to-publish). Those licensing terms will supersede any other terms that the author or any third party may assert apply to any version of the manuscript.

29Please note that you will not receive your proofs until the publishing agreement has been received through our system.

For information regarding our different publishing models please see our <https://www.springernature.com/gp/open-research/transformative-journals> Transformative Journals page. If you have any questions about costs, Open Access requirements, or our legal forms, please contact ASJournals@springernature.com.

[REDACTED]

[REDACTED]

Reviewer #1:

Remarks to the Author:

The authors have done a thorough job in responding to my concerns. I have two comments remaining.

1. The analyses concerning the confounding effect of technical variation are convincing. However, I think these analyses should be added to the manuscript as well - the problem of biological and technical variation being potentially confounded is a real issue in studies interested in transcriptome variability, and given the high expected impact of this paper it would be important to a) make people aware of this issue b) clearly show that it is unlikely to be a worry here

2. Higher levels of canalization in sexual vs. asexual organisms is an interesting finding and could benefit from more discussion. Especially, are the analogous findings from lower level organisms and their germline vs somatic cells?

Reviewer #2:

Remarks to the Author:

I am satisfied with the revisions. The authors have addressed my concerns about terminology used and added READMEs and additional text in figure legends that help the reader understand the data.

Reviewer #3:

30Remarks to the Author:

The authors have addressed all of my comments in a satisfactory manner. Please note the following minor edits:

Line 304: "identifies" should be "identities"

Line 746: abbreviate "Drosophila melanogaster" as "D. melanogaster" since it's already been mentioned before?

Author Rebuttal, first revision:

Response to reviewers:

Reviewer #1:

Remarks to the Author:

The authors have done a thorough job in responding to my concerns. I have two comments remaining.

1. The analyses concerning the confounding effect of technical variation are convincing. However, I think these analyses should be added to the manuscript as well - the problem of biological and technical variation being potentially confounded is a real issue in studies interested in transcriptome variability, and given the high expected impact of this paper it would be important to a) make people aware of this issue b) clearly show that it is unlikely to be a worry here

We have now added this information in the manuscript and provided the details in the Supplementary Information.

2. Higher levels of canalization in sexual vs. workers is an interesting finding and could benefit from more discussion. Especially, are the analogous findings from lower level organisms and their germline vs somatic cells?

Discussing these parallels at length will be beyond the scope of our paper. However, to meet the request, we now extend the final paragraph, where we discuss the similarity across domains of organizational complexity.

Reviewer #2:

Remarks to the Author:

I am satisfied with the revisions. The authors have addressed my concerns about terminology used and added READMEs and addition text in figure legends that help the reader understand the data.

Thank you!

Reviewer #3:

Remarks to the Author:

The authors have addressed all of my comments in a satisfactory manner. Please note the following minor edits:

31Thank you!

Line 304: "identifies" should be "identities"

Corrected.

Line 746: abbreviate "Drosophila melanogaster" as "D. melanogaster" since it's already been mentioned before?

Corrected.

Final Decision Letter:

12th August 2022

Dear Guojie,

We are pleased to inform you that your Article entitled "Canalized gene expression during development mediates caste differentiation in ants", has now been accepted for publication in Nature Ecology & Evolution.

Over the next few weeks, your paper will be copyedited to ensure that it conforms to Nature Ecology and Evolution style. Once your paper is typeset, you will receive an email with a link to choose the appropriate publishing options for your paper and our Author Services team will be in touch regarding any additional information that may be required

You will not receive your proofs until the publishing agreement has been received through our system

Due to the importance of these deadlines, we ask you please us know now whether you will be difficult to contact over the next month. If this is the case, we ask you provide us with the contact information (email, phone and fax) of someone who will be able to check the proofs on your behalf, and who will be available to address any last-minute problems . Once your paper has been scheduled for online publication, the Nature press office will be in touch to confirm the details.

Acceptance of your manuscript is conditional on all authors' agreement with our publication policies (see www.nature.com/authors/policies/index.html). In particular your manuscript must not be published elsewhere and there must be no announcement of the work to any media outlet until the publication date (the day on which it is uploaded onto our web site).

Please note that *Nature Ecology & Evolution* is a Transformative Journal (TJ). Authors may

32publish their research with us through the traditional subscription access route or make their paper immediately open access through payment of an article-processing charge (APC). Authors will not be required to make a final decision about access to their article until it has been accepted. [Find out more about Transformative Journals](https://www.springernature.com/gp/open-research/transformative-journals)

Authors may need to take specific actions to achieve [compliance with funder and institutional open access mandates](https://www.springernature.com/gp/open-research/funding/policy-compliance-faqs). If your research is supported by a funder that requires immediate open access (e.g. according to [Plan S principles](https://www.springernature.com/gp/open-research/plan-s-compliance)) then you should select the gold OA route, and we will direct you to the compliant route where possible. For authors selecting the subscription publication route, the journal's standard licensing terms will need to be accepted, including [self-archiving-and-license-to-publish](https://www.nature.com/nature-portfolio/editorial-policies/self-archiving-and-license-to-publish). Those licensing terms will supersede any other terms that the author or any third party may assert apply to any version of the manuscript.

We welcome the submission of potential cover material (including a short caption of around 40 words) related to your manuscript; suggestions should be sent to Nature Ecology & Evolution as electronic files (the image should be 300 dpi at 210 x 297 mm in either TIFF or JPEG format). Please note that such pictures should be selected more for their aesthetic appeal than for their scientific content, and that colour images work better than black and white or grayscale images. Please do not try to design a cover with the Nature Ecology & Evolution logo etc., and please do not submit composites of images related to your work. I am sure you will understand that we cannot make any promise as to whether any of your suggestions might be selected for the cover of the journal.

You can generate the link yourself when you receive your article DOI by entering it here: http://authors.springernature.com/share.

[REDACTED]

P.S. Click on the following link if you would like to recommend Nature Ecology & Evolution to your librarian <http://www.nature.com/subscriptions/recommend.html#forms>

** Visit the Springer Nature Editorial and Publishing website at www.springernature.com/editorial-and-publishing-jobs for more information about our career opportunities. If you have any questions please click here. **